# Missing Data Imputation by Reducing Mutual Information with Rectified Flows

**Jiahao Yu**[*]
University of Cambridge

**Qizhen Ying**[*]
University of Oxford

**Leyang Wang**[*]
University College London

**Ziyue Jiang**
University of Bristol

**Song Liu**[†]
University of Bristol

## Abstract

This paper introduces a novel iterative method for missing data imputation that sequentially reduces the mutual information between data and the corresponding missingness mask. Inspired by GAN-based approaches that train generators to decrease the predictability of missingness patterns, our method explicitly targets this reduction in mutual information. Specifically, our algorithm iteratively minimizes the KL divergence between the joint distribution of the imputed data and missingness mask, and the product of their marginals from the previous iteration. We show that the optimal imputation under this framework can be achieved by solving an ODE whose velocity field minimizes a rectified flow training objective. We further illustrate that some existing imputation techniques can be interpreted as approximate special cases of our mutual-information-reducing framework. Comprehensive experiments on synthetic and real-world datasets validate the efficacy of our proposed approach, demonstrating its superior imputation performance. Our implementation is available at `https://github.com/yujhml/MIRI-Imputation`.

## 1 Introduction

The problem of missing data, referring to absent components (`NaNs`) in a dataset, can arise in a wide range of scenarios [31, 18]. For example, in a survey dataset, respondents might skip some questions accidentally or intentionally, resulting in missing values. In fMRI data, there can be missing voxels due to incomplete brain coverage and spatial variations in images acquired across subjects [36]. Simple "one-shot" methods replace missing entries with summary statistics, such as the sample mean or median of the observed data. In contrast, approaches like MICE [37], MissForest [34], and HyperImpute [12] employ an iterative, dimension-wise scheme. In each iteration, they model one feature conditional on all others and sample from that model to replace the feature's missing entries, cycling through all features repeatedly until the imputations stabilize. Because their updates are sequential and dimension-wise, these methods are difficult to parallelize and scale poorly to high-dimensional data such as images [3]. Their accuracy is also highly sensitive to the choice of per-variable models [37, 15].

In recent years, research on generative modeling has made huge progress. Since imputation is naturally a data-generating task, generative models have been increasingly applied to impute missing data. [30] proposes to maximize the likelihood of the imputed data using an EM algorithm, where the likelihood function is modeled using a normalizing flow [29, 6]. GAIN [38] leverages Generative Adversarial Nets (GANs) [8] to impute missing components. It trains a generator (a neural network) so that a binary classifier cannot distinguish whether a given component is imputed or not. However, both GANs and normalizing flows come with their own challenges. GANs rely on unstable adversarial

---

[*]Work partially completed at the University of Bristol.

[†]Correspondence to Song Liu <song.liu@bristol.ac.uk>.

39th Conference on Neural Information Processing Systems (NeurIPS 2025).

training and may suffer from mode collapse [16, 40], while normalizing flows require specially designed architectures to ensure their invertibility, limiting their applications.

Currently, diffusion models and *flow-based methods* represent the state of the art in generative modeling techniques [33, 17, 21]. They first train a velocity field according to some criteria. Then, they sample from the target distribution by solving an ODE or SDE using the trained velocity field as the drift. The main challenge to adapt these methods for missing value imputation is that most flow-based methods are designed to transport reference samples to match the target distribution. However, it is unclear how to formulate the missing data imputation problem as a distribution transport problem. Nonetheless, progress has been made in this area. For example, MissDiff [27] learns a score model from partially observed samples, then uses an inpainter [23] to impute missing values.

Recently, [20] observes that GAIN's criterion encourages the imputed dataset to be independent of the missingness pattern. In this paper, we further explore this idea within a more rigorous framework. Our contributions are summarized as follows:

1. We introduce **Mutual Information Reducing Iterations (MIRI)**, a novel framework that provably reduces mutual information between imputed data and their missingness mask when using an optimal imputer;

2. We show that an optimal MIRI imputer can be obtained by solving an ODE whose velocity field is trained by a rectified flow objective, naturally integrating generative modeling;

3. We reveal that several existing imputation methods can be viewed as approximate special cases of the MIRI framework;

4. We demonstrate that our proposed approach achieves promising empirical performance on both tabular and image datasets.

## 2 Background

We now briefly review the missing data imputation problem, Generative Adversarial Imputation Nets (GAIN) [38], and Rectified Flow [21], which form the foundation of our proposed method.

**Notations.** We denote scalars with lowercase letters (e.g., $x, y$) and vectors with bold lowercase letters (e.g., $\mathbf{x}, \mathbf{y}$). Random variables are denoted by uppercase letters (e.g., $X, Y$), and random vectors by bold uppercase letters (e.g., $\mathbf{X}, \mathbf{Y}$). The superscript $t \in \{1, \ldots, T\}$ denotes the iteration count of the algorithm. The subscripts $0, 1$, and $\tau$ denote the continuous time index in an ODE. Given a fixed missingness mask $\mathbf{m}$ and a vector $\mathbf{x}$ of the same dimension, we write $\mathbf{x_m}$ as the subvector of $\mathbf{x}$ whose elements correspond to entries where $m_i = 1$, and $\mathbf{x_{1-m}}$ as the subvector whose elements correspond to entries where $m_i = 0$. Following convention, we refer to $\mathbf{x_m}$ as the "non-missing components" of $\mathbf{x}$ and $\mathbf{x_{1-m}}$ as the "missing components" of $\mathbf{x}$. The notation $A \stackrel{\text{d}}{=} B$ means that the random variables $A$ and $B$ are equal in distribution.

### 2.1 Sequential Imputation

Now, we formally define the missing data imputation problem. The true data vector $\mathbf{X}^* = [X_1^*, \ldots, X_d^*]$ is a random vector taking values on $\mathcal{X} \subset \mathbb{R}^d$, while the missingness mask $\mathbf{M} = [M_1, \ldots, M_d]$ is a random vector in $\{0, 1\}^d$. In the *Missing Completely at Random (MCAR)* [31] setting, we assume that the true data is independent of the missingness mask, i.e., $\mathbf{X}^* \perp\!\!\!\perp \mathbf{M}$ (See Section F for discussions on Missing At Random (MAR) setting). The observed data vector $\tilde{\mathbf{X}} = (\tilde{X}_1, \ldots, \tilde{X}_d)$ is defined as $\tilde{X}_j := \begin{cases} X_j^* & M_j = 1 \\ \texttt{NaN} & M_j = 0 \end{cases}$. In words, the variable $j$ in the observation $\tilde{\mathbf{X}}$ is missing when $M_j = 0$ and is observed when $M_j = 1$. Equivalently,

$$\tilde{\mathbf{X}} = \mathbf{M} \odot \mathbf{X}^* + (1 - \mathbf{M}) \odot \texttt{NaN}, \tag{1}$$

where $\odot$ represents element-wise product. In this paper, we want to construct an imputation vector

$$\mathbf{X}(\mathbf{g}) = \mathbf{M} \odot \tilde{\mathbf{X}} + (1 - \mathbf{M}) \odot \mathbf{g}(\mathbf{Z}_0; \tilde{\mathbf{X}}, \mathbf{M}), \tag{2}$$

where $\mathbf{g} : \mathbb{R}^d \times (\mathbb{R}^d \cup \mathtt{NaN}) \times \{0, 1\}^d \to \mathbb{R}^d$ is called an *imputer*. $\mathbf{Z}_0$ is a random seed of the imputer. Different random seeds $\mathbf{Z}_0$ can lead to different imputations.

In many cases, imputation is an iterative process: Instead of computing $\mathbf{X}$ in a single step, the imputation is improved through iterative updates. This sequential imputation involves the following procedure: for each iteration $t = 1, \ldots, T$, we construct the imputation

$$\mathbf{X}^{(t)}(\mathbf{g}_t) := \mathbf{M} \odot \mathbf{X}^{(t-1)} + (1 - \mathbf{M}) \odot \mathbf{g}_t(\mathbf{Z}_0^{(t)}; \mathbf{X}^{(t-1)}, \mathbf{M}), \tag{3}$$

where $\mathbf{X}^{(t)}$ indicates the imputed data vector at iteration $t$. The initial vector, $\mathbf{X}^{(0)}$ is initialized with any standard imputation technique (e.g., Gaussian noise, mean or median imputation). A schematic of this sequential imputation algorithm is provided in Figure 4 of Appendix A.

The central task of this iterative process is to learn the imputer $\mathbf{g}_t$ at each iteration $t$. In the following sections, we show how a popular missing data imputation scheme can be formulated within this framework and how $\mathbf{g}_t$ can be trained using a GAN-type objective.

## 2.2 Generative Adversarial Imputation Nets (GAIN) [38]

GAIN arises from a natural intuition: if an imputation is perfect, the imputed entries should be indistinguishable from the originally observed ones. Thus, GAIN trains a generator so that a classifier cannot predict the missingness mask $\mathbf{M}$ accurately.

Let $f(m_j, \mathbf{x}, \mathbf{m}_{-j}) \in [0, 1]$ be a probabilistic classifier modeling the conditional probability $\mathbb{P}_{M_j|\mathbf{X}^{(t-1)}, \mathbf{M}_{-j}}$, which is the probability that the $j$-th component is observed given the imputed data from the previous step, $\mathbf{X}^{(t-1)}$, and the rest of the missingness mask, $\mathbf{M}_{-j}$.

For each iteration $t = 1, \ldots, T$, GAIN performs the following steps:

1. Classifier $f_{tj} := \arg\min_f -\mathbb{E}\left[\log f\left(M_j, \mathbf{X}^{(t-1)}, \mathbf{M}_{-j}\right)\right]$, for all $j = 1 \ldots d$.

2. Imputer $\mathbf{g}_t := \arg\max_{\mathbf{g}} - \sum_j \mathbb{E}\left[\log f_{tj}\left(M_j, \mathbf{X}^{(t)}(\mathbf{g}), \mathbf{M}_{-j}\right)\right] - \lambda(\text{reconstruction error})$.

3. Update $\mathbf{X}^{(t)} \overset{\text{impute}}{\leftarrow} \mathbf{X}^{(t)}(\mathbf{g}_t)$, $t \leftarrow t + 1$.

Here, $\mathbf{X}^{(t)}(\mathbf{g})$ is the imputed vector from the imputer $\mathbf{g}$ as defined in (3). At iteration $t$, we first train classifiers $f_{tj}$ by *minimizing* the cross-entropy loss. Subsequently, we train the imputer $\mathbf{g}_t$, parameterized as a neural network, to reduce the classifiers' predictive accuracy by *maximizing* the sum of cross-entropy losses across all $j$, while simultaneously minimizing the reconstruction error.

In practice, the minimization and maximization steps are performed via alternating gradient updates, leading to adversarial training. We observe that GAIN works well in practice. However, the adversarial training is known to be difficult and may suffer from mode collapses [16, 40]. Moreover, balancing the reconstruction error and the cross entropy loss requires careful tuning of the hyperparameter $\lambda$.

Recently, it was observed that the generator training of GAIN can be viewed as a process of *breaking the dependency* between $\mathbf{M}$ and $\mathbf{X}^{(t)}(\mathbf{g})$ [20]. Indeed, if $\mathbf{X}^{(t)}(\mathbf{g}) \perp\!\!\!\perp \mathbf{M}$, meaning $\mathbf{X}^{(t)}(\mathbf{g})$ is not predictive of $\mathbf{M}$ at all, then the cross-entropy loss is maximized. In this paper, we rigorously explore this idea of imputation by dependency reduction. Guided by this principle, our algorithm explicitly aims to reduce the dependency between $\mathbf{M}$ and $\mathbf{X}^{(t)}(\mathbf{g})$.

## 2.3 Flow-based Sampling and Rectified Flow

As an alternative to GANs, flow-based generative models have attracted significant interest in recent years. These algorithms first train a "velocity field" over time $\tau \in [0, 1]$ according to some loss function, and then generate samples by solving the ODE or SDE using the learned velocity field as the drift. In some cases, the loss function is a simple least-squares loss, leading to a training routine simpler and more stable than GAN. Methods in this category include Score-based generative models [32, 33], Rectified Flow [21], Flow Matching [17], Diffusion Schrödinger Bridge [5], among others.

Rectified flow is one of the simplest flow-based generative models. Given two random vectors $\mathbf{X}_0$ (from a reference distribution) and $\mathbf{X}_1$ (from a target distribution), the velocity field $\mathbf{v}^*$ is trained by

---
**Algorithm 1** MIRI with Rectified Flow (Single Imputation)

---
**Require:** Paired, i.i.d. observations and masks $\{(\tilde{\mathbf{X}}, \mathbf{M})\}$, Maximum Iterations $T$, Maximum SGD steps $N$, Batch Size $B$, Optimizer `SGD_update` and an ODE solver `ODE_Solver`.
1: $(\mathbf{X}^{(0)}, \mathbf{M}) \leftarrow$ `initial_impute`$(\tilde{\mathbf{X}}, \mathbf{M})$.
2: **for** $t = 1$ to $T$ **do**
3:     Initialize a neural network model $\mathbf{v}$.
4:     **for** $n = 1$ to $N$ **do**
5:         Sample batch $\{(\mathbf{X}_0^{(j)}, \mathbf{M}_0^{(j)})\}_{j=1}^{B} \sim \mathbb{P}_{\mathbf{X}^{(t-1)}, \mathbf{M}_0}$,
6:         Sample batch $\{\mathbf{X}_1^{(j)}\}_{j=1}^{B} \sim \mathbb{P}_{\mathbf{X}^{(t-1)}}$, and time indices $\{\tau_j\}_{j=1}^{B} \sim \mathrm{Uniform}(0, 1)$
7:         $\forall j, \mathbf{X}_{\tau_j}^{(j)} \leftarrow (1 - \tau_j)\mathbf{X}_0^{(j)} + \tau_j \mathbf{X}_1^{(j)}, \mathbf{Y}^{(j)} \leftarrow \mathbf{X}_1^{(j)} - \mathbf{X}_0^{(j)}$
8:         $\mathbf{v}_t \leftarrow$ `SGD_update` $\left( \nabla_{\mathbf{v}} \sum_j \left\| \mathbf{Y}^{(j)} - \mathbf{v}(\mathbf{X}_{\tau_j}^{(j)}, \mathbf{M}_0^{(j)} \odot \mathbf{X}_0^{(j)}, \mathbf{M}_0^{(j)} \odot \mathbf{X}_1^{(j)}, \mathbf{M}_0^{(j)}, \tau_j) \right\|^2 \right)$
9:     **end for**
10:    $\mathbf{X}^{(t)} \leftarrow \mathbf{M} \odot \mathbf{X}^{(t-1)} + (1 - \mathbf{M}) \odot$ `ODE_Solver`$((1 - \mathbf{M}) \odot \mathbf{v}_t, \mathbf{X}^{(t-1)}, \mathbf{M})$
11: **end for**
12: **return** $\{\mathbf{X}^{(T)}\}$

---

minimizing the following objective:

$$\mathbf{v}^* = \arg\min_{\mathbf{v}} \int_0^1 \mathbb{E}\left[\|\mathbf{X}_1 - \mathbf{X}_0 - \mathbf{v}(\mathbf{X}_\tau, \tau)\|^2\right] d\tau, \tag{4}$$

where $\mathbf{X}_\tau$ is the linear interpolation between $\mathbf{X}_0$ and $\mathbf{X}_1$, defined as $\mathbf{X}_\tau = \tau\mathbf{X}_1 + (1 - \tau)\mathbf{X}_0$ for $\tau \in [0, 1]$. Samples are then generated by solving the ODE:

$$\frac{d\mathbf{Z}_\tau}{d\tau} = \mathbf{v}^*(\mathbf{Z}_\tau, \tau). \tag{5}$$

One can prove that, if $\mathbf{Z}_0 \stackrel{d}{=} \mathbf{X}_0$, then $\mathbf{Z}_\tau \stackrel{d}{=} \mathbf{X}_\tau$ for all $\tau \in [0, 1]$. This "marginal-preserving property" guarantees that solving the ODE from $\tau = 0$ to $\tau = 1$ transports samples from the reference distribution to the target distribution.

## 3 Reducing Dependency with Mutual Information

In this section, we introduce the MIRI imputation framework, obtain the optimal imputer under this framework using rectified flow, and theoretically justify its validity under the Missing Completely at Random (MCAR) setting. The validity for the Missing at Random (MAR) scenario is in Appendix F.

### 3.1 Mutual Information Reducing Iterations (MIRI)

We study the problem of training an imputer $\mathbf{g}$ to *reduce the dependency* between the imputed sample $\mathbf{X}(\mathbf{g})$ and missingness mask $\mathbf{M}$. The approach adopted by [20] is to minimize the mutual information between them, i.e., the optimal $\mathbf{g}$ is given as:

$$\mathbf{g} \in \arg\min_{\mathbf{g}} \mathrm{I}(\mathbf{X}(\mathbf{g}); \mathbf{M}) = \arg\min_{\mathbf{g}} \mathrm{D}[\mathbb{P}_{\mathbf{X}(\mathbf{g}), \mathbf{M}} \| \mathbb{P}_{\mathbf{X}(\mathbf{g})} \otimes \mathbb{P}_{\mathbf{M}}], \tag{6}$$

where $\mathrm{D}[\mathbb{P}\|\mathbb{Q}]$ is the Kullback-Leibler (KL) divergence of probability distributions $\mathbb{Q}$ from $\mathbb{P}$.

However, the mutual information is not directly computable as we do not have access to the true distribution $\mathbb{P}_{\mathbf{X}(\mathbf{g}), \mathbf{M}}$ and $\mathbb{P}_{\mathbf{X}(\mathbf{g})} \otimes \mathbb{P}_{\mathbf{M}}$. A potential solution is to transform this optimization problem into a bi-level min-max adversarial optimization problem: Fixing $\mathbf{g}$, it is possible to estimate the mutual information using samples from $\mathbb{P}_{\mathbf{X}(\mathbf{g}), \mathbf{M}}$ and $\mathbb{P}_{\mathbf{X}(\mathbf{g})} \otimes \mathbb{P}_{\mathbf{M}}$ with Mutual Information Neural Estimation (MINE) [1], which *maximizes* the Donsker-Varadhan lower bound [7] to approximate the KL divergence. After that, we can *minimize* the estimated mutual information with respect to $\mathbf{g}$. This process is repeated until convergence. If one solves both optimization problems one gradient step at a time, this algorithm is the classic adversarial training that GAN is known for.

*Contrary* to the suggestion in [20], the mutual information in (6) *cannot* be minimized simply by directly applying forward or reverse KL Wasserstein Gradient Flow (WGF). While WGF can minimize a KL divergence $\mathrm{D}[\mathbb{P}\|\mathbb{Q}]$ when the distribution to be optimized is either $\mathbb{P}$ or $\mathbb{Q}$, the

distribution of interest in our setting ($\mathbb{P}_{\mathbf{X(g)}}$) appears simultaneously in both $\mathbb{P}$ and $\mathbb{Q}$. As a result, neither the forward nor reverse KL formulation of WGF is directly applicable. Therefore, while the WGF procedure proposed in [20] is conceptually appealing, it does not correctly minimize the mutual information objective as formulated in (6).

Motivated by the limitation of both adversarial training and WGF, we propose a sequential imputation algorithm that reduces the mutual information, called Mutual Information Reducing Iterations (MIRI). Instead of targeting the unknown marginal distribution $\mathbb{P}_{\mathbf{X(g)}}$, we use the imputed data distribution from the *previous* iteration, $\mathbf{X}^{(t-1)}$, as a stable target. Let us denote the joint probability of the pair $(\mathbf{X}^{(t)}(\mathbf{g}), \mathbf{M})$ as $\mathbb{P}_{\mathbf{X}^{(t)}(\mathbf{g}),\mathbf{M}}$ and the product measure of $\mathbf{X}^{(t-1)}$ and $\mathbf{M}$ as $\mathbb{P}_{\mathbf{X}^{(t-1)}} \otimes \mathbb{P}_{\mathbf{M}}$.

For each iteration $t = 1 \ldots T$, the algorithm performs the following steps:

1. **Find the Optimal Imputer**: $\mathbf{g}_t \in \arg\min_{\mathbf{g}} \mathrm{D} \left[ \mathbb{P}_{\mathbf{X}^{(t)}(\mathbf{g}),\mathbf{M}} \| \mathbb{P}_{\mathbf{X}^{(t-1)}} \otimes \mathbb{P}_{\mathbf{M}} \right]$.

2. **Update Data**: $\mathbf{X}^{(t)} \overset{\text{impute}}{\leftarrow} \mathbf{X}^{(t)}(\mathbf{g}_t), t \leftarrow t + 1$.

We can confirm that MIRI does indeed reduce the mutual information:

**Proposition 1.** *The mutual information between $\mathbf{X}^{(t)}$ and $\mathbf{M}$ is non-increasing after each iteration.*

See Appendix B for the proof. In Section 4.1, we show that the GAIN algorithm can be viewed as an approximate implementation of the above MIRI algorithm.

Moreover, we show that minimizing the KL divergence in the first step of MIRI has a sufficient and necessary condition:

**Proposition 2.** *Let $p_{\mathbf{g}}(\mathbf{x}, \mathbf{m})$ be the density of $(\mathbf{X}^{(t)}(\mathbf{g}_t), \mathbf{M})$ and $q(\mathbf{x})$ be the density of $\mathbf{X}^{(t-1)}$. $\mathbf{g} \in \arg\min_{\mathbf{g}} \mathrm{D} \left[ \mathbb{P}_{\mathbf{X}^{(t)}(\mathbf{g}),\mathbf{M}} \| \mathbb{P}_{\mathbf{X}^{(t-1)}} \otimes \mathbb{P}_{\mathbf{M}} \right]$ if and only if*

$$p_{\mathbf{g}}(\mathbf{x}_{1-\mathbf{m}}|\mathbf{x}_{\mathbf{m}}, \mathbf{m}) = q(\mathbf{x}_{1-\mathbf{m}}|\mathbf{x}_{\mathbf{m}}), \forall \mathbf{x}, \mathbf{m}. \tag{7}$$

The proof can be found in the Appendix E. This result shows that the optimal $\mathbf{g}$ should sample from the conditional distribution of the missing components given the non-missing components according to the marginal distribution of $\mathbf{X}^{(t-1)}$, the imputed data in the previous iteration. (7) inspires us to construct a flow-based generative model with a target distribution $q(\mathbf{x}_{1-\mathbf{m}}|\mathbf{x}_{\mathbf{m}})$.

As we discuss in Section 4.2 and 4.3, (7) is also the key design principle behind many classic and contemporary data imputation algorithms.

## 3.2   Imputation by Rectified Flow

In this section, we show how to construct an optimal imputer for MIRI using *an imputation ODE* obtained via rectified flow training. The optimality condition of Proposition 2 requires an imputer to sample from target distribution $q(\mathbf{x}_{1-\mathbf{m}}|\mathbf{x}_{\mathbf{m}})$. The standard diffusion models are restrictive, as they are constrained to a fixed Gaussian reference. We therefore use rectified flow to transport samples from the current imputation (reference) to the target distribution.

We now zoom in on Step 1 of the MIRI algorithm at iteration $t$.

Let $(\mathbf{X}_1, \mathbf{M}_1)$ be a pair of random vectors drawn from the product measure $\mathbb{P}_{\mathbf{X}^{(t-1)}} \otimes \mathbb{P}_{\mathbf{M}}$ and $(\mathbf{X}_0, \mathbf{M}_0)$ from the joint distribution $\mathbb{P}_{\mathbf{X}^{(t-1)},\mathbf{M}}$. The pairs are drawn such that $\mathbf{X}_1 \perp\!\!\!\perp (\mathbf{X}_0, \mathbf{M}_0)$. $\mathbf{X}_\tau$ is defined as the linear interpolation of $\mathbf{X}_0$ and $\mathbf{X}_1$, following the same construction as in Section 2.3.

Consider the following rectified flow training objective function:

$$\mathbf{v}^* := \arg\min_{\mathbf{v}} \int_0^1 \mathbb{E} \left[ \| \mathbf{X}_1 - \mathbf{X}_0 - \mathbf{v}(\mathbf{X}_{\tau,1-\mathbf{M}_0}, \mathbf{X}_{0,\mathbf{M}_0}, \mathbf{X}_{1,\mathbf{M}_0}, \mathbf{M}_0, \tau) \|^2 \right] \mathrm{d}\tau, \tag{8}$$

where $\mathbf{X}_{0,\mathbf{M}_0} := \mathbf{M}_0 \odot \mathbf{X}_0$ and others are defined similarly.

We can define an *imputation process* by using the optimal velocity field $\mathbf{v}^*$. Given a partially observed vector where the missing entries are padded with zeros, i.e., $[\mathbf{0}, \mathbf{x}_{\mathbf{m}}]$, the imputation process dynamics are governed by the following ODE:

$$\frac{d\mathbf{Z}_\tau}{d\tau} = (1 - \mathbf{m}) \odot \mathbf{v}^*(\mathbf{Z}_\tau, [\mathbf{0}, \mathbf{x}_{\mathbf{m}}], [\mathbf{0}, \mathbf{x}_{\mathbf{m}}], \mathbf{m}, \tau), \tag{9}$$

with initial condition

$$\mathbf{Z}_0 \sim \mathbb{P}_{\mathbf{X}_{0,1-\mathbf{m}} \,\big|\, \{\mathbf{X}_{0,\mathbf{m}}=[\mathbf{0},\mathbf{x_m}],\, \mathbf{M}_0=\mathbf{m}\}}. \tag{10}$$

We define the imputer $\mathbf{g}^*$ as the solution to the imputation ODE (9) evaluated at terminal time $\tau = 1$:

$$\mathbf{g}^*(\mathbf{Z}_0, \mathbf{x_m}, \mathbf{m}) := \mathbf{Z}_1. \tag{11}$$

We can show that, $\mathbf{g}^*$ *is an optimal imputer* to the MIRI algorithm that we introduced in Section 3.1.

**Theorem 1.** $\mathbf{g}^*$ *defined in* (11) *is an optimal imputer in the sense that*

$$\mathbf{g}^* \in \arg\min_{\mathbf{g}} \mathrm{D} \left[ \mathbb{P}_{\mathbf{X}^{(t)}(\mathbf{g}),\mathbf{M}} \| \mathbb{P}_{\mathbf{X}^{(t-1)}} \otimes \mathbb{P}_{\mathbf{M}} \right]. \tag{12}$$

The main technical challenge of the proof is to show that the ODE (9) can indeed establish a "marginal-preserving" process from $\mathbb{P}_{\mathbf{X}^{(t-1)}|\mathbf{M}}$ to $\mathbb{P}_{\mathbf{X}^{(t-1)}}$. This is different from the classic proof since both our training objective and ODE are different from the regular rectified flow. See Appendix C for the proof. With an imputer $\mathbf{g}^*$, we can proceed to the second step of the MIRI algorithm.

### 3.3 Initial Condition

To run the imputation process, we require at least a sample of $\mathbf{X}_{0,1-\mathbf{m}} \,\big|\, \{\mathbf{X}_{0,\mathbf{m}} = \mathbf{x_m}, \mathbf{M} = \mathbf{m}\}$. By the definition of $\mathbf{X}_0$, this is equivalent to sampling from $\mathbf{X}_{1-\mathbf{m}}^{(t-1)}|\{\mathbf{X}_{\mathbf{m}}^{(t-1)} = \mathbf{x_m}, \mathbf{M} = \mathbf{m}\}$. This poses no difficulty, since at each iteration $t$, we maintain access to the joint sample $(\mathbf{X}^{(t-1)}, \mathbf{M})$, from which we can extract the relevant conditional sample by slicing. However, in the first iteration, we need samples from $\mathbf{X}_{1-\mathbf{m}}^{(0)}|\{\mathbf{X}_{\mathbf{m}}^{(0)} = \mathbf{x_m}, \mathbf{M} = \mathbf{m}\}$ to kickstart the MIRI algorithm.

One can sample from any distribution for those initial samples and in our experiment, we simply draw independent samples from the normal $N(0,1)$ or uniform $U(0,1)$ to fill out each missing component. Alternatively, one can use another imputation algorithm to suggest initial samples.

### 3.4 Practical Implementation

Algorithm 1 presents the full MIRI procedure on a finite sample set $(\tilde{\mathbf{X}}, \mathbf{M})$. Note that the superscript $(t)$ represents iteration count while $(j)$ is the sample index in a batch. We model $\mathbf{v}_t$ with a neural network trained via stochastic gradient descent. The function `initial_impute` replaces `NaN`s with initial guesses.

According to the requirement in Section 3.2, we need to ensure $(\mathbf{X}_0, \mathbf{M}_0) \perp\!\!\!\perp \mathbf{X}_1$. In practice, we propose to sample a batch $\{\mathbf{X}_{0,j}, \mathbf{M}_{0,j}\}_{i=1}^B$ from the paired dataset and sample a batch $\{\mathbf{X}_{1,j}\}_{i=1}^B$ from a *shuffled set* of $\{\mathbf{X}_{0,j}\}$ to weaken their dependency. Although, in this case, $\mathbf{X}_1$ is still not independent of $\mathbf{X}_0$ and $\mathbf{M}_0$, we observe that this strategy works well in practice.

## 4 MIRI and Other Imputation Algorithms

In this section, we show how MIRI is related to other existing imputation algorithms.

### 4.1 GAIN

Let $q(\mathbf{x}, \mathbf{m})$ be the density function of the joint distribution of $(\mathbf{X}^{(t-1)}, \mathbf{M})$. Suppose the discriminator is well-trained in the first step in the GAIN algorithm, i.e., $f(m_j, \mathbf{x}, \mathbf{m}_{-j}) \approx q(m_j|\mathbf{x}, \mathbf{m}_{-j})$, then the generator training in the second step (without the reconstruction error) is

$$\mathbf{g}_t = \arg\min_{\mathbf{g}} \mathbb{E} \left[ \sum_j \log q(M_j|\mathbf{X}_{\mathbf{g}}^{(t)}, \mathbf{M}_{-j}) \right] = \arg\min_{\mathbf{g}} \mathbb{E} \left[ \log \prod_j q(M_j|\mathbf{X}_{\mathbf{g}}^{(t)}, \mathbf{M}_{-j}). \right] \tag{13}$$

Using the pseudo-likelihood approximation [2], $\prod_j q(M_j|\mathbf{X}_{\mathbf{g}}^{(t)}, \mathbf{M}_{-j}) \approx q(\mathbf{M}|\mathbf{X}_{\mathbf{g}}^{(t)})$, thus:

$$\mathbf{g}_t \approx \arg\min_{\mathbf{g}} \mathbb{E} \left[ \log q(\mathbf{M}|\mathbf{X}_{\mathbf{g}}^{(t)}) \right] = \arg\min_{\mathbf{g}} \mathbb{E} \left[ \log \frac{q(\mathbf{M}, \mathbf{X}_{\mathbf{g}}^{(t)})}{q(\mathbf{X}_{\mathbf{g}}^{(t)})p(\mathbf{M})} \right]. \tag{14}$$

By Gibbs' inequality,

$$\mathbb{E}\left[\log\frac{q(\mathbf{M},\mathbf{X}_{\mathbf{g}}^{(t)})}{q(\mathbf{X}_{\mathbf{g}}^{(t)})p(\mathbf{M})}\right] \leq \mathbb{E}\left[\log\frac{p_{\mathbf{g}}(\mathbf{M},\mathbf{X}_{\mathbf{g}}^{(t)})}{q(\mathbf{X}_{\mathbf{g}}^{(t)})p(\mathbf{M})}\right] = \mathrm{D}\left[\mathbb{P}_{\mathbf{X}^{(t)}(\mathbf{g}),\mathbf{M}}\|\mathbb{P}_{\mathbf{X}^{(t-1)}}\otimes\mathbb{P}_{\mathbf{M}}\right], \qquad (15)$$

thus, we can see that GAIN learns its imputer $\mathbf{g}_t$ by approximately minimizing *a lower bound* of $\mathrm{D}\left[\mathbb{P}_{\mathbf{X}^{(t)}(\mathbf{g}),\mathbf{M}}\|\mathbb{P}_{\mathbf{X}^{(t-1)}}\otimes\mathbb{P}_{\mathbf{M}}\right]$.

There are two approximations made in the above argument: the pseudo-likelihood approximation and the Gibbs' inequality. First, since GAIN only performs one gradient step update to the imputer $\mathbf{g}_t$ at each iteration, the gap between $\mathrm{D}\left[\mathbb{P}_{\mathbf{X}^{(t)}(\mathbf{g}),\mathbf{M}}\|\mathbb{P}_{\mathbf{X}^{(t-1)}}\otimes\mathbb{P}_{\mathbf{M}}\right]$ and its lower bound may not be large. Second, the pseudo-likelihood approximation can be a good approximation, for example, when the correlation among $M_j$ is not too strong. The discriminator training of GAIN resembles a dimension-wise strategy that has been widely used in Ising model estimation [28, 25]. Note that the optimization in (14) is fundamentally intractable due to the high-dimensional density $q(\mathbf{m}|\mathbf{x}^{(t)}(\mathbf{g}))$. $\mathbf{m}$ is a binary variable defined on $\{0,1\}^d$, thus, the density does not have a tractable normalizing constant. This further restricts us to the dimension-wise pseudo-likelihood approach in GAIN algorithm.

## 4.2  Round-Robin Approaches: MICE, HyperImpute, etc.

Recall that Proposition 2 states that minimizing the KL divergence $\mathrm{D}\left[\mathbb{P}_{\mathbf{X}^{(t)}(\mathbf{g}),\mathbf{M}}\|\mathbb{P}_{\mathbf{X}^{(t-1)}}\otimes\mathbb{P}_{\mathbf{M}}\right]$ is equivalent to choosing a $\mathbf{g}$ such that $p_{\mathbf{g}}(\mathbf{x}_{1-\mathbf{m}}|\mathbf{x}_{\mathbf{m}},\mathbf{m}) = q(\mathbf{x}_{1-\mathbf{m}}|\mathbf{x}_{\mathbf{m}})$, where $p_{\mathbf{g}}(\mathbf{x},\mathbf{m})$ is the density of $(\mathbf{X}^{(t)}(\mathbf{g}),\mathbf{M})$ and $q(\mathbf{x})$ is the density of $\mathbf{X}^{(t-1)}$. One can simply build an imputer that samples from $q(\mathbf{x}_{1-\mathbf{m}}|\mathbf{x}_{\mathbf{m}})$ to impute the missing components. Then, by construction, $p_{\mathbf{g}}(\mathbf{x}_{1-\mathbf{m}}|\mathbf{x}_{\mathbf{m}},\mathbf{m}) = q(\mathbf{x}_{1-\mathbf{m}}|\mathbf{x}_{\mathbf{m}})$.

This algorithm has two issues: First, $\mathbf{m}$ changes for every sample, so we need to learn a different imputer for every sample. Second, sampling from the conditional probability distribution $q(\mathbf{x}_{1-\mathbf{m}}|\mathbf{x}_{\mathbf{m}})$ is not trivial as the dimension of $\mathbf{x}_{1-\mathbf{m}}$ can still be high.

In principle, we can learn the *joint probability* $q$ then sample from the conditional distributions $q(\mathbf{x}_{1-\mathbf{m}}|\mathbf{x}_{\mathbf{m}})$ for any fixed $\mathbf{m}$ (see the next section). However, learning a high dimensional joint probability density function $q$ is not trivial. Thus, we adopt the pseudo-likelihood approximation again: factorize $q(\mathbf{x})$ over dimension-wise conditional probabilities and learn $q(x_j|\mathbf{x}_{-j})$ for every $j$. This dimension-wise conditional probability learning scheme not only makes the learning of the joint distribution easier but also solves the second issue as we can perform *Gibbs sampling*, one missing component at a time. This pseudo-likelihood approximation + Gibbs sampling strategy gives rise to the round-robin algorithms such as MICE and HyperImpute.

Let $f_j$ be the density model of $q(x_j|\mathbf{x}_{-j})$, then the "MIRI MICE" performs the following algorithm. For $t = 1, \ldots, T$:

1. $\forall j$, learn $f_j$ to approximate $q(x_j|\mathbf{x}_{-j})$.
2. Let $\mathbf{g}_t$ be a Gibbs sampler for $q(\mathbf{x}_{1-\mathbf{m}}|\mathbf{x}_{\mathbf{m}})$ using $\{f_j\}$.
3. $\mathbf{X}^{(t)} \overset{\text{impute}}{\longleftarrow} \mathbf{X}^{(t)}(\mathbf{g}_t), t \leftarrow t+1$.

In practice, MICE interlaces first step and the later two steps, meaning a Gibbs sampling step and imputation is immediately performed after learning each $f_j$. However, the sequential nature of Gibbs sampling prevent us from parallelizing this procedure for high-dimensional samples.

## 4.3  Contemporary Approaches: Learn Jointly, Sample/Inpaint Conditionally

Traditionally, learning a high-dimensional joint distribution $q(\mathbf{x})$ is intractable due to its normalising constant. Recently, it was realized that one can learn a joint model $q(\mathbf{x})$ without dealing with the normalising constant. [35] first estimates the joint model $q$ via Score Matching [11] or Noise Contrastive Estimation [10] , then sample from $q(\mathbf{x}_{1-\mathbf{m}}|\mathbf{x}_{\mathbf{m}})$ via importance sampling. This method works well when $\mathbf{x}_{1-\mathbf{m}}$ is in a relatively low-dimensional space. [4] noticed that given a joint model $q$, this conditional sampling problem could be solved using Stein Variational Gradient Descent (SVGD) [19]. However, SVGD utilizes properties of RKHS function, and require kernel computations. Therefore, does not scale to high-dimensional datasets.

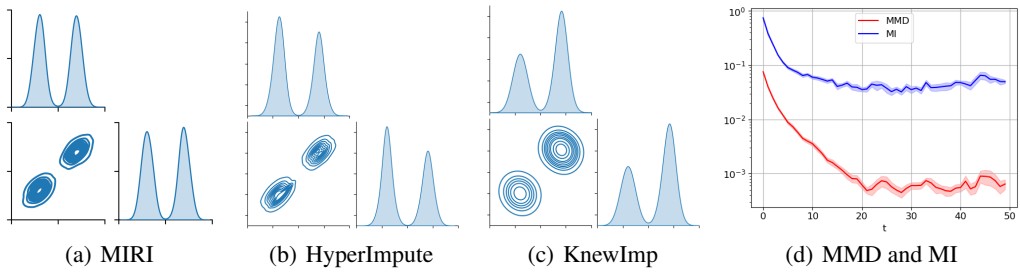

|   |   |   |   |
|---|---|---|---|
| (a) MIRI | (b) HyperImpute | (c) KnewImp | (d) MMD and MI |

Figure 1: Pairwise density plots of the imputed data and MMD/MI of MIRI.

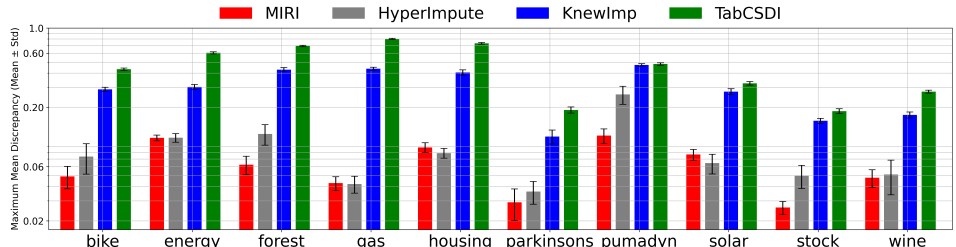

Figure 2: MMD on UCI datasets with 60% data missing. The lower the better.

In recent years, it is also realized that one can "inpaint" samples given a diffusion model trained on the joint samples [23]. This idea gives rise to DiffPuter [39], an imputer that trains a joint diffusion model to sample from $q$. Then use an inpainter to impute missing values given the observed vector $\mathbf{x_m}$. However, an inpainter is not a standard backward process of diffusion dynamics, thus in general, it is not guaranteed to generate samples from $q(\mathbf{x_{1-m}}|\mathbf{x_m})$.

## 5 Experiments

This section evaluates MIRI's performance on a variety of imputation tasks. Details on experimental settings and implementations can be found in Appendix G. Additional results can be found in Appendix H. To ensure fair comparisons, we match model architectures across all methods wherever possible. For example, if our method uses CNN, we use the same architectures for other neural-network-based methods as well.

Imputation methods aim to either predict the conditional mean $\mathbb{E}[\mathbf{X_{1-m}}|\mathbf{X_m}]$, evaluated by metrics like MAE/RMSE, or learn to sample from the conditional distribution $\mathbb{P}_{\mathbf{X_{1-m}}|\mathbf{X_m}}$, evaluated by the distributional fidelity. Our method, MIRI, is designed for the latter. Thus, we mostly evaluate the performance by using metrics that measure the distributional fidelity (such as MMD [9]).

### 5.1 Toy Example: Assessing Imputation Quality Using MMD and Mutual Information

In this section, we showcase the performance of the proposed MIRI imputer on a toy example that is a mixture of 2-dimensional Gaussian: $\mathbf{X}^* \sim 0.5\mathcal{N}([-2,-2], 0.5^2\mathbf{I}) + 0.5\mathcal{N}([2,2], 0.5^2\mathbf{I})$. The missingness mask is generated as $\mathbf{M} \sim \prod_{j=1}^{d} \text{Bernoulli}_{M_j}(0.7)$, i.e., 30% of the data is Missing Completely at Random. We draw 6000 i.i.d. samples from $\mathbb{P}_{\mathbf{X}^*}$ and $\mathbb{P}_{\mathbf{M}}$ to create missing observations. The imputed samples are visualized in the pairwise density plots in Figure 1. We choose HyperImpute [12] and KnewImp [4] as comparisons. It can be seen that MIRI not only captures the bimodal structure of the true data but also preserves the proportion of the two modes. On the other hand, HyperImpute and KnewImp fail to recover the correct proportion of the two modes. We also show performance metrics (MAE, RMSE, MMD) with various other imputation algorithms (such as MissDiff, GAIN, TabCSDI [42]) in Table 5 in the Appendix H.1. Finally, in Figure 1(d), we show that both MIRI's MMD and MI decrease over iterations, proving that MI between the imputed data and the missingness mask is a good criterion for assessing how well the imputed data recovers $\mathbb{P}_{\mathbf{X}^*}$.

Table 2: **Quantitative results on CIFAR-10.** Methods are evaluated at three levels of missingness (20%, 40%, 60%) using FID, PSNR, and SSIM. The best results are highlighted in bold.

| Method | 20% Missingness | | | 40% Missingness | | | 60% Missingness | | |
|---|---|---|---|---|---|---|---|---|---|
| | FID ($\downarrow$) | PSNR ($\uparrow$) | SSIM ($\uparrow$) | FID ($\downarrow$) | PSNR ($\uparrow$) | SSIM ($\uparrow$) | FID ($\downarrow$) | PSNR ($\uparrow$) | SSIM ($\uparrow$) |
| GAIN [38] | 164.11 | 21.21 | 0.7803 | 281.62 | 16.20 | 0.5576 | 285.53 | 11.99 | 0.2933 |
| KnewImp [4] | 153.09 | 18.84 | 0.6463 | 193.68 | 15.81 | 0.4740 | 264.40 | 14.04 | 0.3317 |
| MissDiff [27] | 90.51 | 22.29 | 0.7702 | 129.84 | 19.65 | 0.6648 | 197.91 | 16.78 | 0.4989 |
| HyperImpute [12] | 8.92 | **34.09** | **0.9750** | 65.01 | 23.22 | 0.7931 | 130.36 | 20.17 | 0.6533 |
| **MIRI (Ours*)** | **6.01** | 32.29 | 0.9736 | **27.53** | **27.14** | **0.9126** | **68.58** | **23.22** | **0.8063** |

## 5.2 Real-World Tabular Imputation: UCI Regression Benchmarks

We evaluate MIRI on 10 standard UCI regression benchmarks [13]. We select these datasets for their diversity in sample size and dimensionality (see Table 4 in Appendix G.1). Following [26], we simulate missingness under MCAR/MAR/MNAR mechanisms. We simulate 20%, 40%, and 60% missingness for MCAR and MNAR, and 40% and 80% for MAR. For the MAR mechanism, the missingness mask is conditioned on a randomly selected 50% of the variables. In the main paper, we compare our method with HyperImpute, KnewImp, TabCSDI, GAIN, MIWAE [24], and TDM [41].

Table 1: **Aggregated MMD rankings for UCI (40% missingness) across 10 datasets.** Lower is better.

| Method | MCAR | MAR | MNAR |
|---|---|---|---|
| TabCSDI [42] | 6.4 | 6.4 | 6.1 |
| GAIN [38] | 6.2 | 6.2 | 5.9 |
| TDM [41] | 4.5 | 4.4 | 4.2 |
| KnewImp [4] | 4.1 | 4.2 | 3.8 |
| MIWAE [24] | 3.0 | 2.9 | 2.8 |
| HyperImpute [12] | 1.6 | **1.5** | 1.6 |
| **MIRI (Ours*)** | **1.4** | 2.0 | **1.3** |

Figure 2 reports the average MMD over 10 trials (error bars denote one standard deviation). As shown, even in the 60% missing data cases, MIRI outperforms or performs comparably to all competitors. We aggregate performance using mean rank across 10 datasets. As shown in Table 1 for the 40% missingness setting, MIRI attains the best mean rank under MCAR and MNAR settings and is highly competitive under MAR setting.

Comprehensive results, including performance across all specified missingness levels and comparisons with an extended set of baseline methods, are deferred to Appendix H.2.

## 5.3 Real-World Image Imputation: CIFAR-10 and CelebA Benchmarks

We assess MIRI on high-dimensional image datasets.

First, on CIFAR-10 [14] (32×32 RGB), we train using only 5 000 samples (<10% of the full set) with 60% of pixels randomly removed (all RGB channels jointly). Figure 3(a) compares MIRI's 60% missingness reconstructions against generative approaches (GAIN, KnewImp, MissDiff). Table 2 reports FID, PSNR, and SSIM at 20%, 40%, and 60% missingness, showing MIRI strongly outperforms the best generative alternatives across this range, even in challenging cases.

Table 3: **Accuracy of 10-class CIFAR-10 classification on imputed data at varying missingness.** Higher is better.

| Method | 20% | 40% | 60% |
|---|---|---|---|
| KnewImp [4] | 0.267 | 0.137 | 0.106 |
| GAIN [38] | 0.337 | 0.133 | 0.096 |
| MissDiff [27] | 0.486 | 0.269 | 0.152 |
| HyperImpute [12] | 0.804 | 0.405 | 0.212 |
| **MIRI (Ours*)** | **0.812** | **0.525** | **0.364** |

See Appendix H.3 for additional results, and Table 3 for downstream classification results.

Second, we test MIRI on higher-resolution CelebA [22] (64×64 RGB) with 5 000 samples, using a mechanism that masks each RGB channel independently. Figure 3(b) again substantiates MIRI's strong imputation performance.

# 6 Limitations and Future Works

Despite the competitive results, MIRI has several limitations that open avenues for future work.

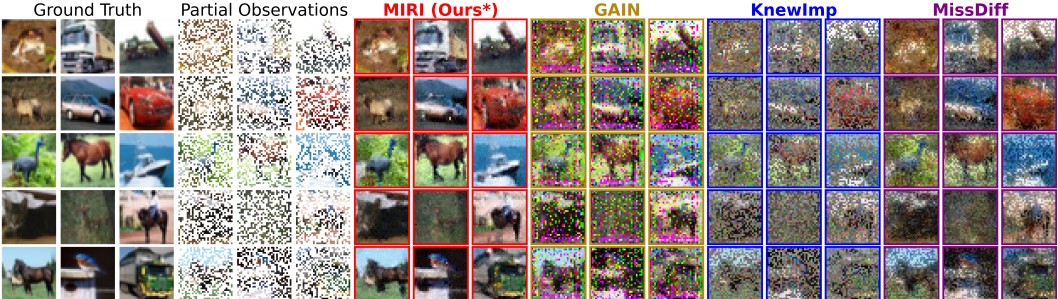

(a) 15 uncurated 32×32 CIFAR-10 images and their imputations. Pixels are removed from *all RGB channels*.

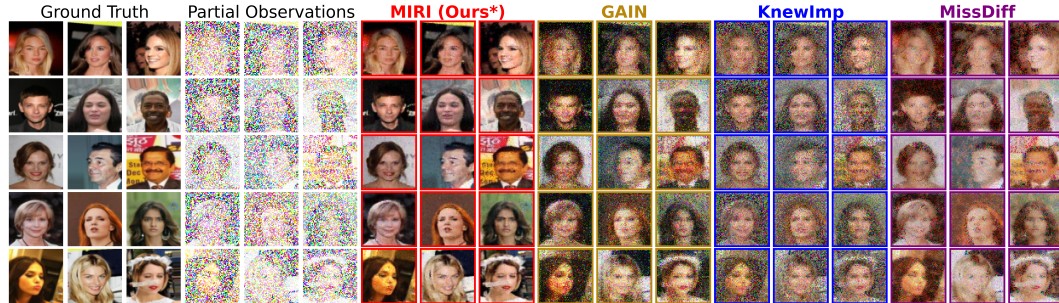

(b) 15 uncurated 64×64 CelebA images and their imputations. Pixels are removed from *each RGB channel independently*.

Figure 3: Comparison of imputed samples on CIFAR-10 and CelebA with 60% of missingness.

A primary limitation is computational intensity. Each MIRI iteration requires running a full rectified flow algorithm, including training a velocity field and solving the corresponding ODE. One interesting future work is to incorporate recent distillation methods to accelerate sample generation.

Our Theorem 1 is a population-level result, meaning it assumes the optimal velocity field $\mathbf{v}^*$ is obtained and the imputation ODE is solved exactly. It does not consider errors due to finite sample approximation, optimisation and inaccurate ODE solver. Extending our theoretical results to the finite-sample optimal $\hat{\mathbf{v}}$ and the approximate imputation ODE solver (like the Euler method) is another avenue for future works.

A more fundamental limitation is that our objective, which minimizes the mutual information between the data and the mask, is not designed for Missing Not at Random (MNAR) scenarios where the missingness mechanism depends on the true values. This is a common challenge for many modern imputation frameworks, and extending our method to handle such cases is a crucial direction.

Finally, MIRI could be extended to a self-supervised framework for time-series. By training on artificially masked, fully-observed sequences, we could learn a general-purpose imputation vector field. The key advantage here is flexibility: because MIRI's vector fields are agnostic to missingness patterns, this approach could yield a single, pre-trained foundation model for robust, general-purpose time-series imputation and forecasting.

## 7   Conclusion

In this work, we introduced *Mutual Information Reducing Iterations* (MIRI), a framework that reduces data-mask dependency by iteratively minimizing $\mathrm{D}[\mathbb{P}_{\mathbf{X}^{(t)}(\mathbf{g}),\mathbf{M}} \| \mathbb{P}_{\mathbf{X}^{(t-1)}} \otimes \mathbb{P}_{\mathbf{M}}]$. We proved that MIRI decreases mutual information $\mathrm{I}(\mathbf{X};\mathbf{M})$. The optimal imputer is constructed by solving an ODE trained with a rectified flow objective. We also showed that existing imputation methods such as GAIN are approximate special cases of this approach. Empirically, MIRI also demonstrated strong distributional fidelity across synthetic, tabular, and image benchmarks.

## Acknowledgments

We thank the four anonymous reviewers and the area chair for their insightful comments and suggestions. SL thanks Josh Givens, Sam Power, Katarzyna Reluga and other colleagues at Bristol Mathematics for helpful discussions. Partial funding for this work was provided by the University of Bristol School of Mathematics Summer Research Bursary 2024; JY was funded by the Alumni Foundation, and LW by the Heilbronn Institute for Mathematical Research. JY also acknowledges support from the Scholar Award and the ED Davies Travel Fund, provided by the Neural Information Processing Foundation and the Fitzwilliam College, University of Cambridge, respectively. Partial computational resources were provided by the Advanced Computing Research Centre at the University of Bristol (https://www.bristol.ac.uk/acrc/).

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

## A  Illustration of Sequential Imputation

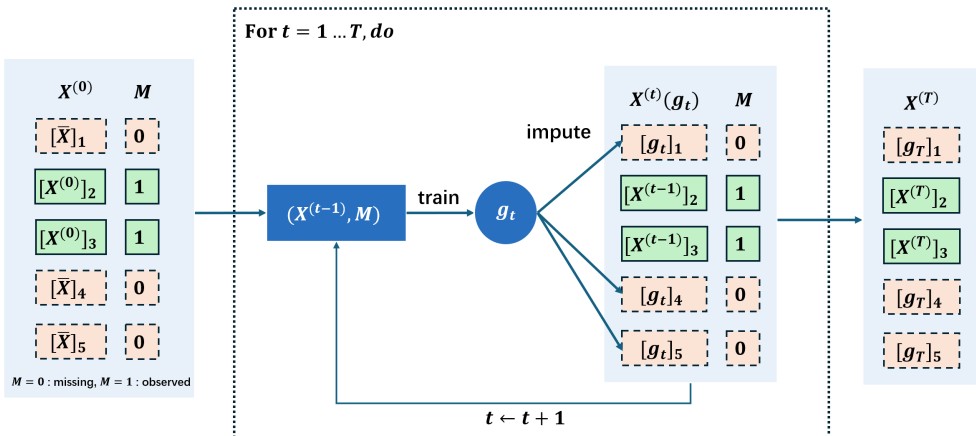

Figure 4: Schematic of the sequential imputation algorithm. The process begins with an initial imputation, $\mathbf{X}^{(0)}$, and iteratively refines the missing values over $T$ steps.

Figure 4 illustrates the sequential imputation framework, where a model is iteratively trained to refine estimates for missing data.

## B  Proof of Proposition 1

*Proof.* Consider the difference between the mutual information at iteration $t$ and $t-1$:

$$\mathrm{D}\left[\mathbb{P}^{(t)}_{\mathbf{X},\mathbf{M}}\|\mathbb{P}^{(t)}_{\mathbf{X}}\otimes\mathbb{P}_{\mathbf{M}}\right] - \mathrm{D}\left[\mathbb{P}_{\mathbf{X}^{(t-1)},\mathbf{M}}\|\mathbb{P}_{\mathbf{X}^{(t-1)}}\otimes\mathbb{P}_{\mathbf{M}}\right]$$

$$=\underbrace{\mathrm{D}\left[\mathbb{P}^{(t)}_{\mathbf{X},\mathbf{M}}\|\mathbb{P}^{(t)}_{\mathbf{X}}\otimes\mathbb{P}_{\mathbf{M}}\right] - \mathrm{D}\left[\mathbb{P}^{(t)}_{\mathbf{X},\mathbf{M}}\|\mathbb{P}_{\mathbf{X}^{(t-1)}}\otimes\mathbb{P}_{\mathbf{M}}\right]}_{A} + \underbrace{\mathrm{D}\left[\mathbb{P}^{(t)}_{\mathbf{X},\mathbf{M}}\|\mathbb{P}_{\mathbf{X}^{(t-1)}}\otimes\mathbb{P}_{\mathbf{M}}\right] - \mathrm{D}\left[\mathbb{P}_{\mathbf{X}^{(t-1)},\mathbf{M}}\|\mathbb{P}_{\mathbf{X}^{(t-1)}}\otimes\mathbb{P}_{\mathbf{M}}\right]}_{B}.$$

$$(16)$$

Since $\mathrm{D}\left[\mathbb{P}_{\mathbf{X}^{(t)}(\mathbf{g}),\mathbf{M}}\|\mathbb{P}_{\mathbf{X}^{(t-1)}}\otimes\mathbb{P}_{\mathbf{M}}\right]$ is minimized at $\mathbf{g}_t$ and $\mathbf{X}^{(t)}=\mathbf{X}^{(t)}(\mathbf{g}_t)$ by definition,

$$\mathrm{D}\left[\mathbb{P}^{(t)}_{\mathbf{X},\mathbf{M}}\|\mathbb{P}_{\mathbf{X}^{(t-1)}}\otimes\mathbb{P}_{\mathbf{M}}\right] \leq \mathrm{D}\left[\mathbb{P}_{\mathbf{X}^{(t)}(\mathbf{g}),\mathbf{M}}\|\mathbb{P}_{\mathbf{X}^{(t-1)}}\otimes\mathbb{P}_{\mathbf{M}}\right], \forall\mathbf{g}. \qquad (17)$$

Moreover, $\mathrm{D}\left[\mathbb{P}_{\mathbf{X}^{(t-1)},\mathbf{M}}\|\mathbb{P}_{\mathbf{X}^{(t-1)}}\otimes\mathbb{P}_{\mathbf{M}}\right] = \mathrm{D}\left[\mathbb{P}_{\mathbf{X}^{(t)}(\mathbf{g}),\mathbf{M}}\|\mathbb{P}_{\mathbf{X}^{(t-1)}}\otimes\mathbb{P}_{\mathbf{M}}\right]|_{\mathbf{g}=\mathrm{Identity}}$, so $B$ is non-positive. To show $A$ is non-positive, we write out the KL-divergences in $A$. Let $p(\mathbf{x},\mathbf{m})$ be the density of $(\mathbf{X}^{(t)},\mathbf{M})$ and $q(\mathbf{x})$ be the density of $\mathbf{X}^{(t-1)}$.

$$\mathrm{D}\left[\mathbb{P}^{(t)}_{\mathbf{X},\mathbf{M}}\|\mathbb{P}^{(t)}_{\mathbf{X}}\otimes\mathbb{P}_{\mathbf{M}}\right] - \mathrm{D}\left[\mathbb{P}^{(t)}_{\mathbf{X},\mathbf{M}}\|\mathbb{P}_{\mathbf{X}^{(t-1)}}\otimes\mathbb{P}_{\mathbf{M}}\right] \qquad (18)$$

$$=\mathbb{E}\left[\log\frac{p(\mathbf{X}^{(t)},\mathbf{M})}{p(\mathbf{X}^{(t)})p(\mathbf{M})}\right] - \mathbb{E}\left[\log\frac{p(\mathbf{X}^{(t)},\mathbf{M})}{q(\mathbf{X}^{(t)})p(\mathbf{M})}\right] \qquad (19)$$

$$=-\mathbb{E}\left[\log p(\mathbf{X}^{(t)})\right] + \mathbb{E}\left[\log q(\mathbf{X}^{(t)})\right] \leq 0, \qquad (20)$$

where the last inequality is due to the Gibbs inequality.

Thus, $A+B \leq 0$ as desired. $\qquad\square$

## C  Proof of Theorem 1

*Proof.* First, we introduce the following lemma, based on the marginal preserving property of rectified flow.

**Lemma 1.** *The imputation ODE with the initial condition given in* (10)*, i.e.,*

$$\mathbf{Z}_0 \overset{d}{=} \mathbf{X}_{0,1-\mathbf{m}} | \left\{ \mathbf{X}_{0,\mathbf{m}} = \mathbf{x_m}, \mathbf{M}_0 = \mathbf{m} \right\} \tag{21}$$

*has a solution at the terminal time* $\tau = 1$, $\mathbf{Z}_1 \overset{d}{=} \mathbf{X}_{1,1-\mathbf{m}} | \left\{ \mathbf{X}_{1,\mathbf{m}} = \mathbf{x_m} \right\}$, *for every fixed* $\mathbf{x}, \mathbf{m}$.

See the Appendix D for the proof.

Now we derive the main result. Lemma 1 shows that under the initial condition, the solution $\mathbf{Z}_1$ is

$$\begin{aligned}
\mathbf{Z}_1 &\overset{d}{=} \mathbf{X}_{1,1-\mathbf{m}} | \left\{ \mathbf{X}_{1,\mathbf{m}} = \mathbf{x_m} \right\} \\
&= \mathbf{X}_{1-\mathbf{m}}^{(t-1)} | \left\{ \mathbf{X}_{\mathbf{m}}^{(t-1)} = \mathbf{x_m} \right\}
\end{aligned} \tag{22}$$

Moreover, $\mathbf{Z}_1$ is also our imputer $\mathbf{g}_t$, so by construction (9),

$$\mathbf{Z}_1 \overset{d}{=} \mathbf{X}_{\mathbf{g}^*,\mathbf{m}}^{(t)} | \left\{ \mathbf{X}_{1-\mathbf{m}}^{(t-1)} = \mathbf{x}_{1-\mathbf{m}}, \mathbf{M}_0 = \mathbf{m} \right\}. \tag{23}$$

Combining (23) and (22), we obtain the following equality

$$\mathbb{P}_{\mathbf{X}_{\mathbf{g}^*,1-\mathbf{m}}^{(t)} | \mathbf{X}_{\mathbf{m}}^{(t-1)} = \mathbf{x_m}, \mathbf{M}=\mathbf{m}} = \mathbb{P}_{\mathbf{X}_{1-\mathbf{m}}^{(t-1)} | \mathbf{X}_{\mathbf{m}}^{(t-1)} = \mathbf{x_m}}. \tag{24}$$

The imputation ODE does not change observed components where $m_j = 1$, so $\mathbf{X}_{\mathbf{m}}^{(t-1)} = \mathbf{X}_{\mathbf{m}}^{(t)}$. We have

$$\mathbb{P}_{\mathbf{X}_{\mathbf{g}^*,1-\mathbf{m}}^{(t)} | \mathbf{X}_{\mathbf{m}}^{(t-1)} = \mathbf{x_m}, \mathbf{M}=\mathbf{m}} = \mathbb{P}_{\mathbf{X}_{\mathbf{g}^*,1-\mathbf{m}}^{(t)} | \mathbf{X}_{\mathbf{m}}^{(t)} = \mathbf{x_m}, \mathbf{M}=\mathbf{m}}. \tag{25}$$

(24) and (25) imply that $p_{\mathbf{g}^*}(\cdot | \mathbf{x_m}, \mathbf{m}) = q(\cdot | \mathbf{x_m})$. According to Proposition 2, we have shown that $\mathbf{g}^*$ is an optimal imputer that minimizes $\mathrm{D}\left[ \mathbb{P}_{\mathbf{X}^{(t)}(\mathbf{g}),\mathbf{M}} \| \mathbb{P}_{\mathbf{X}^{(t-1)}} \otimes \mathbb{P}_{\mathbf{M}} \right]$. $\qquad\square$

## D Proof of Lemma 1

**Definition 1.** *For fixed values* $\mathbf{y}, \mathbf{u}, \mathbf{v}, \mathbf{m}$*, the event* $\Xi$ *is defined as*

$$\Xi(\mathbf{u}, \mathbf{w}, \mathbf{m}) = \{ \mathbf{X}_{0,\mathbf{m}} = \mathbf{u}, \mathbf{X}_{1,\mathbf{m}} = \mathbf{w}, \mathbf{M}_0 = \mathbf{m} \}. \tag{26}$$

*and the optimal solution of* (8) *is*

$$\mathbf{v}_{\mathbf{m}}^*(\mathbf{y}, \mathbf{u}, \mathbf{w}, \mathbf{m}, \tau) := \mathbb{E}\left[ \dot{\mathbf{X}}_{\tau,1-\mathbf{m}} \,\middle|\, \mathbf{X}_{\tau,1-\mathbf{m}} = \mathbf{y}, \Xi(\mathbf{u}, \mathbf{w}, \mathbf{m}) \right]. \tag{27}$$

*We denote the density of random variable* $\mathbf{X}_{\tau,1-\mathbf{m}} | \Xi(\mathbf{u}, \mathbf{w}, \mathbf{m})$ *as* $p_\tau(\mathbf{y} | \mathbf{u}, \mathbf{w}, \mathbf{m})$.

**Lemma 2.** $p_0(\mathbf{y} | \mathbf{u}, \mathbf{w}, \mathbf{m}) = p_0(\mathbf{y} | \mathbf{u}, \mathbf{m})$ *and* $p_1(\mathbf{y} | \mathbf{u}, \mathbf{w}, \mathbf{m}) = p_1(\mathbf{y} | \mathbf{w})$.

*Proof.* We can see that due to the independence $\mathbf{X}_1 \perp\!\!\!\perp (\mathbf{X}_0, \mathbf{M}_0)$,

$$\mathbf{X}_{0,1-\mathbf{m}} | \mathbf{X}_{0,\mathbf{m}}, \mathbf{X}_{1,\mathbf{m}}, \mathbf{M}_0 = \mathbf{X}_{0,1-\mathbf{m}} | \mathbf{X}_{0,\mathbf{m}}, \mathbf{M}_0, \tag{28}$$

hence the first equality holds. What's more,

$$\mathbf{X}_{1,1-\mathbf{m}} | \mathbf{X}_{0,\mathbf{m}}, \mathbf{X}_{1,\mathbf{m}}, \mathbf{M}_0 = \mathbf{X}_{1,1-\mathbf{m}} | \mathbf{X}_{1,\mathbf{m}}, \tag{29}$$

hence the second equality holds. $\qquad\square$

**Lemma 3.** *The density function* $p_\tau(\mathbf{y} | \mathbf{u}, \mathbf{w}, \mathbf{m})$ *satisfies the continuity equation*

$$\partial_\tau p_\tau(\mathbf{y} | \mathbf{u}, \mathbf{w}, \mathbf{m}) + \nabla_{\mathbf{y}} \cdot \left( p_\tau(\mathbf{y} | \mathbf{u}, \mathbf{w}, \mathbf{m}) \, \mathbf{v}^*(\mathbf{y}, \mathbf{u}, \mathbf{w}, \mathbf{m}, \tau) \right) = 0 \tag{30}$$

*Proof.* Consider the following expectation:

$$\mathbb{E}\big[h(\mathbf{X}_{\tau,1-\mathbf{m}})|\Xi(\mathbf{u},\mathbf{w},\mathbf{m})\big] = \int_{\mathbb{R}^{d_{1-\mathbf{m}}}} h(\mathbf{y})\,p_\tau(\mathbf{y}|\mathbf{u},\mathbf{w},\mathbf{m})\,\mathrm{d}\mathbf{y} \tag{31}$$

whose time derivative is

$$\partial_\tau \mathbb{E}\big[h(\mathbf{X}_{\tau,1-\mathbf{m}})|\Xi(\mathbf{u},\mathbf{w},\mathbf{m})\big] = \int h(\mathbf{y})\,\partial_\tau p_\tau(\mathbf{y}|\mathbf{u},\mathbf{w},\mathbf{m})\,\mathrm{d}\mathbf{y}. \tag{32}$$

On the other hand, we can see that $\mathbf{X}_\tau$ is a deterministic function of $\mathbf{X}_0$ and $\mathbf{X}_1$, thus using the law of unconscious statistician:

$$\partial_\tau \mathbb{E}\big[h(\mathbf{X}_{\tau,1-\mathbf{m}})|\Xi(\mathbf{u},\mathbf{w},\mathbf{m})\big] \tag{33}$$
$$=\partial_\tau \mathbb{E}_{\mathbf{X}_0,\mathbf{X}_1}\big[h(\mathbf{X}_{\tau,1-\mathbf{m}})|\Xi(\mathbf{u},\mathbf{w},\mathbf{m})\big] \tag{34}$$
$$=\mathbb{E}_{\mathbf{X}_0,\mathbf{X}_1}\big[\partial_\tau h(\mathbf{X}_{\tau,1-\mathbf{m}})|\Xi(\mathbf{u},\mathbf{w},\mathbf{m})\big] \tag{35}$$
$$=\mathbb{E}\big[\partial_\tau h(\mathbf{X}_{\tau,1-\mathbf{m}})|\Xi(\mathbf{u},\mathbf{w},\mathbf{m})\big]. \tag{36}$$

Expanding the previous expression using the chain rule,

$$\partial_\tau \mathbb{E}\big[h(\mathbf{X}_{\tau,1-\mathbf{m}})|\Xi(\mathbf{u},\mathbf{w},\mathbf{m})\big] \tag{37}$$
$$=\mathbb{E}\big[\partial_\tau h(\mathbf{X}_{\tau,1-\mathbf{m}})|\Xi(\mathbf{u},\mathbf{w},\mathbf{m})\big] \tag{38}$$
$$=\mathbb{E}\Big[\nabla h(\mathbf{X}_{\tau,1-\mathbf{m}})^\top \dot{\mathbf{X}}_{\tau,1-\mathbf{m}}\,\Big|\,\Xi(\mathbf{u},\mathbf{w},\mathbf{m})\Big] \tag{39}$$
$$=\mathbb{E}\Big[\mathbb{E}\big[\nabla h(\mathbf{X}_{\tau,1-\mathbf{m}})^\top \dot{\mathbf{X}}_{\tau,1-\mathbf{m}}\,\big|\mathbf{X}_{\tau,1-\mathbf{m}};\Xi(\mathbf{u},\mathbf{w},\mathbf{m})\big]\,\Big|\,\Xi(\mathbf{u},\mathbf{w},\mathbf{m})\Big] \tag{40}$$
$$=\mathbb{E}\Big[\nabla h(\mathbf{X}_{\tau,1-\mathbf{m}})^\top \mathbb{E}\big[\dot{\mathbf{X}}_{\tau,1-\mathbf{m}}\,\big|\,\mathbf{X}_{\tau,1-\mathbf{m}};\Xi(\mathbf{u},\mathbf{w},\mathbf{m})\big]\,\Big|\,\Xi(\mathbf{u},\mathbf{w},\mathbf{m})\Big] \tag{41}$$
$$=\int \nabla h(\mathbf{y})^\top \mathbf{v}^*(\mathbf{y},\mathbf{u},\mathbf{w},\mathbf{m},\tau)\,p_\tau(\mathbf{y}|\mathbf{u},\mathbf{w},\mathbf{m})\,\mathrm{d}\mathbf{y} \tag{42}$$
$$=-\int h(\mathbf{y})\,\nabla\!\cdot\!\Big(\mathbf{v}^*(\mathbf{y},\mathbf{u},\mathbf{w},\mathbf{m},\tau)\,p_\tau(\mathbf{y}|\mathbf{u},\mathbf{w},\mathbf{m})\Big)\,\mathrm{d}\mathbf{y} \tag{43}$$

where the last equality by performing integration by parts on the right-hand side (with respect to $\mathbf{y}$).

Now we equate the two representations ((32) and (43)) for $\partial_\tau \mathbb{E}[h(\mathbf{X}_{\tau,1-\mathbf{m}})|\Xi(\mathbf{u},\mathbf{w},\mathbf{m})]$:

$$\int h(\mathbf{y})\,\partial_\tau p_\tau(\mathbf{y}|\mathbf{u},\mathbf{w},\mathbf{m})\,\mathrm{d}\mathbf{y} = -\int h(\mathbf{y})\,\nabla\!\cdot\!\Big(\mathbf{v}^*(\mathbf{y},\mathbf{u},\mathbf{w},\mathbf{m},\tau)\,p_\tau(\mathbf{y}|\mathbf{u},\mathbf{w},\mathbf{m})\Big)\,\mathrm{d}\mathbf{y}. \tag{44}$$

Since this equality holds for every smooth, compactly supported test function $h(\mathbf{y})$, the fundamental lemma of the calculus of variations implies that,

$$\partial_\tau p_\tau(\mathbf{y}|\mathbf{u},\mathbf{w},\mathbf{m}) + \nabla\!\cdot\!\Big(\mathbf{v}^*(\mathbf{y},\mathbf{u},\mathbf{w},\mathbf{m},\tau)\,p_\tau(\mathbf{y}|\mathbf{u},\mathbf{w},\mathbf{m})\Big) = 0. \tag{45}$$

We can see that $\mathbf{v}^*$ satisfies the continuity equation for a time-varying density function $p_\tau$. $\qquad\square$

We can see that with the initial condition

$$\mathbf{Z}_0 = \mathbf{X}_{0,1-\mathbf{m}}|\Xi(\mathbf{u},\mathbf{w},\mathbf{m}),$$

Lemma 3 says

$$\mathbf{Z}_1 = \mathbf{X}_{1,1-\mathbf{m}}|\Xi(\mathbf{u},\mathbf{w},\mathbf{m}).$$

Using Lemma 2, we simplify $\mathbf{Z}_0$ and $\mathbf{Z}_1$ as

$$\mathbf{Z}_0 = \mathbf{X}_{0,1-\mathbf{m}}|\{\mathbf{X}_{0,\mathbf{m}} = \mathbf{u}, \mathbf{M}_0 = \mathbf{m}\}, \quad \mathbf{Z}_1 = \mathbf{X}_{1,1-\mathbf{m}}|\{\mathbf{X}_{1,\mathbf{m}} = \mathbf{w}\}.$$

Letting $\mathbf{u} = \mathbf{w} = \mathbf{x_m}$, we obtain the desired results in Lemma 1.

# E    Proof of Proposition 2

*Proof.* Since the sequential imputer (3) never changes the observed part $\mathbf{X}_\mathbf{m}^{(t)}$ nor the missing mask $\mathbf{M}$, we have

$$p_\mathbf{g}(\mathbf{x}, \mathbf{m}) = p_\mathbf{g}(\mathbf{x}_{1-\mathbf{m}}|\mathbf{x}_\mathbf{m}, \mathbf{m})p(\mathbf{x}_\mathbf{m}, \mathbf{m}), \tag{46}$$

where the marginal density $p(\mathbf{x}_\mathbf{m}, \mathbf{m})$ does not depend on $\mathbf{g}$. Now we show that the KL divergence only depends on $\mathbf{g}$ through the KL divergence $\mathrm{D}[p_\mathbf{g}(\cdot|\mathbf{x}_{1-\mathbf{m}}, \mathbf{m})|q(\cdot|\mathbf{x}_{1-\mathbf{m}})]$.

$$\mathrm{D}\left[\mathbb{P}_{\mathbf{X}^{(t)}(\mathbf{g}),\mathbf{M}}\|\mathbb{P}_{\mathbf{X}^{(t-1)}}\otimes\mathbb{P}_\mathbf{M}\right] \tag{47}$$

$$=\mathbb{E}_{(\mathbf{x},\mathbf{m})\sim(\mathbf{X}_\mathbf{g}^{(t)},\mathbf{M})}\left[\log\frac{p_\mathbf{g}(\mathbf{x}, \mathbf{m})}{q(\mathbf{x})p(\mathbf{m})}\right] \tag{48}$$

$$=\mathbb{E}_{(\mathbf{x},\mathbf{m})\sim(\mathbf{X}_\mathbf{g}^{(t)},\mathbf{M})}\left[\log\frac{p_\mathbf{g}(\mathbf{x}_{1-\mathbf{m}}, \mathbf{x}_\mathbf{m}, \mathbf{m})}{q(\mathbf{x}_{1-\mathbf{m}}, \mathbf{x}_\mathbf{m})p(\mathbf{m})}\right] \tag{49}$$

$$=\mathbb{E}_{(\mathbf{x},\mathbf{m})\sim(\mathbf{X}_\mathbf{g}^{(t)},\mathbf{M})}\left[\log\frac{p_\mathbf{g}(\mathbf{x}_{1-\mathbf{m}}|\mathbf{x}_\mathbf{m}, \mathbf{m})}{q(\mathbf{x}_{1-\mathbf{m}}|\mathbf{x}_\mathbf{m})}\right] + \mathbb{E}_{(\mathbf{x},\mathbf{m})\sim(\mathbf{X}_\mathbf{g}^{(t)},\mathbf{M})}\left[\log\frac{p(\mathbf{x}_\mathbf{m}, \mathbf{m})}{q(\mathbf{x}_\mathbf{m})p(\mathbf{m})}\right] \tag{50}$$

$$= \sum_{\mathbf{m}\in\{0,1\}^d} p(\mathbf{m})\mathbb{E}_{\mathbf{x}\sim\mathbf{X}_\mathbf{g}^{(t)}|\mathbf{M}=\mathbf{m}}\left[\log\frac{p_\mathbf{g}(\mathbf{x}_{1-\mathbf{m}}|\mathbf{x}_\mathbf{m}, \mathbf{m})}{q(\mathbf{x}_{1-\mathbf{m}}|\mathbf{x}_\mathbf{m})}\right] + \mathrm{const.} \tag{51}$$

$$= \sum_{\mathbf{m}\in\{0,1\}^d} p(\mathbf{m})\underbrace{\int p(\mathbf{x}_\mathbf{m}|\mathbf{m})\mathrm{D}[p_\mathbf{g}(\mathbf{x}_{1-\mathbf{m}}|\mathbf{x}_\mathbf{m}, \mathbf{m})\|q(\mathbf{x}_{1-\mathbf{m}}|\mathbf{x}_\mathbf{m})]\mathrm{d}\mathbf{x}_\mathbf{m}}_{A_\mathbf{g}} + \mathrm{const.}, \tag{52}$$

where const. does not depend on $\mathbf{g}$. The constant in (51) is due to the factorization (46):

$$\mathbb{E}_{(\mathbf{x},\mathbf{m})\sim(\mathbf{X}_\mathbf{g}^{(t)},\mathbf{M})}\left[\log\frac{p(\mathbf{x}_\mathbf{m}, \mathbf{m})}{q(\mathbf{x}_\mathbf{m})p(\mathbf{m})}\right] = \mathbb{E}_{p(\mathbf{x}_\mathbf{m},\mathbf{m})}\mathbb{E}_{p_\mathbf{g}(\mathbf{x}_{1-\mathbf{m}}|\mathbf{x}_\mathbf{m},\mathbf{m})}\left[\log\frac{p(\mathbf{x}_\mathbf{m}, \mathbf{m})}{q(\mathbf{x}_\mathbf{m})p(\mathbf{m})}\right] = \mathrm{const.} \tag{53}$$

Assuming $p(\mathbf{x}_\mathbf{m}|\mathbf{m})$ is positive everywhere, $A_\mathbf{g}$ is minimized if and only if

$$\mathrm{D}[p_\mathbf{g}(\mathbf{x}_{1-\mathbf{m}}|\mathbf{x}_\mathbf{m}, \mathbf{m})|q(\mathbf{x}_{1-\mathbf{m}}|\mathbf{x}_\mathbf{m})] = 0. \tag{54}$$

The KL divergence $\mathrm{D}[p_\mathbf{g}(\mathbf{x}_{1-\mathbf{m}}|\mathbf{x}_\mathbf{m}, \mathbf{m})\|q(\mathbf{x}_{1-\mathbf{m}}|\mathbf{x}_\mathbf{m})]$ is zero if and only if

$$p_\mathbf{g}(\mathbf{x}_{1-\mathbf{m}}|\mathbf{x}_\mathbf{m}, \mathbf{m}) = q(\mathbf{x}_{1-\mathbf{m}}|\mathbf{x}_\mathbf{m}). \tag{55}$$

Thus, if $p(\mathbf{m})$ is strictly positive, $\mathrm{D}\left[\mathbb{P}_{\mathbf{X}^{(t)}(\mathbf{g}),\mathbf{M}}\|\mathbb{P}_{\mathbf{X}^{(t-1)}}\otimes\mathbb{P}_\mathbf{M}\right]$ is minimized if and only if $p_\mathbf{g}(\mathbf{x}_{1-\mathbf{m}}|\mathbf{x}_\mathbf{m}, \mathbf{m}) = q(\mathbf{x}_{1-\mathbf{m}}|\mathbf{x}_\mathbf{m})$.

$\square$

# F    Validity of MIRI under MAR Setting

Recall that we denote the observed components of a vector $\mathbf{X}$ as $\mathbf{X}_\mathbf{M}$ and the missing components as $\mathbf{X}_{1-\mathbf{M}}$. Under the Missing at Random (MAR) assumption, the goal is to minimize the conditional mutual information:

$$\mathrm{I}[\mathbf{X}_{1-\mathbf{M}}(\mathbf{g}); \mathbf{M}|\mathbf{X}_\mathbf{M}] = \mathbb{E}_{(\mathbf{x},\mathbf{m})\sim(\mathbf{X},\mathbf{M})}\left[\log\frac{p_\mathbf{g}(\mathbf{x}_{1-\mathbf{m}}, \mathbf{m}|\mathbf{x}_\mathbf{m})}{p_\mathbf{g}(\mathbf{x}_{1-\mathbf{m}}|\mathbf{x}_\mathbf{m})p(\mathbf{m}|\mathbf{x}_\mathbf{m})}\right]. \tag{56}$$

Following a similar argument to the proof of Proposition 1, we can define an iterative algorithm to reduce this conditional mutual information. At each iteration $t$, we find the optimal imputer $\mathbf{g}_t$ by solving:

$$\mathbf{g}_t \in \arg\min_\mathbf{g} \mathbb{E}_{(\mathbf{x},\mathbf{m})\sim(\mathbf{X}_\mathbf{g}^{(t)},\mathbf{M})}\left[\log\frac{p_\mathbf{g}(\mathbf{x}_{1-\mathbf{m}}, \mathbf{m}|\mathbf{x}_\mathbf{m})}{q(\mathbf{x}_{1-\mathbf{m}}|\mathbf{x}_\mathbf{m})q(\mathbf{m}|\mathbf{x}_\mathbf{m})}\right], \tag{57}$$

where $p_\mathbf{g}$ is the density corresponding to the distribution of $(\mathbf{X}_\mathbf{g}^{(t)}, \mathbf{M})$, and $q$ is the density of $\mathbf{X}^{(t-1)}$ from the previous iteration.

We now show that $p_{\mathbf{g}}(\mathbf{x}_{1-\mathbf{m}}|\mathbf{m}, \mathbf{x_m}) = q(\mathbf{x}_{1-\mathbf{m}}|\mathbf{x_m})$ is a sufficient condition for $\mathbf{g}$ to be optimal. Analogous to the proof of Proposition 2, we can rewrite the objective function. Since the missingness mechanism is fixed and does not depend on the imputer $\mathbf{g}$, we have $p(\mathbf{m}|\mathbf{x_m}) = q(\mathbf{m}|\mathbf{x_m})$. The objective then simplifies:

$$\mathbb{E}_{(\mathbf{x},\mathbf{m})\sim(\mathbf{X}_{\mathbf{g}}^{(t)},\mathbf{M})}\left[\log\frac{p_{\mathbf{g}}(\mathbf{x}_{1-\mathbf{m}},\mathbf{m}|\mathbf{x_m})}{q(\mathbf{x}_{1-\mathbf{m}}|\mathbf{x_m})q(\mathbf{m}|\mathbf{x_m})}\right] \tag{58}$$

$$=\mathbb{E}_{(\mathbf{x},\mathbf{m})\sim(\mathbf{X}_{\mathbf{g}}^{(t)},\mathbf{M})}\left[\log\frac{p_{\mathbf{g}}(\mathbf{x}_{1-\mathbf{m}}|\mathbf{m},\mathbf{x_m})p(\mathbf{m}|\mathbf{x_m})}{q(\mathbf{x}_{1-\mathbf{m}}|\mathbf{x_m})q(\mathbf{m}|\mathbf{x_m})}\right] \tag{59}$$

$$=\mathbb{E}_{(\mathbf{x},\mathbf{m})\sim(\mathbf{X}_{\mathbf{g}}^{(t)},\mathbf{M})}\left[\log\frac{p_{\mathbf{g}}(\mathbf{x}_{1-\mathbf{m}}|\mathbf{m},\mathbf{x_m})}{q(\mathbf{x}_{1-\mathbf{m}}|\mathbf{x_m})}\right] \tag{60}$$

$$=\mathbb{E}_{(\mathbf{m},\mathbf{x_m})\sim(\mathbf{M},\mathbf{X_M})}\mathbb{E}_{\mathbf{x}_{1-\mathbf{m}}\sim p_{\mathbf{g}}(\cdot|\mathbf{m},\mathbf{x_m})}\left[\log\frac{p_{\mathbf{g}}(\mathbf{x}_{1-\mathbf{m}}|\mathbf{m},\mathbf{x_m})}{q(\mathbf{x}_{1-\mathbf{m}}|\mathbf{x_m})}\right] \tag{61}$$

$$=\mathbb{E}_{(\mathbf{m},\mathbf{x_m})\sim(\mathbf{M},\mathbf{X_M})}D[p_{\mathbf{g}}(\cdot|\mathbf{m},\mathbf{x_m})\|q(\cdot|\mathbf{x_m})]. \tag{62}$$

The inner expectation is the KL divergence between the conditional distributions $p_{\mathbf{g}}(\mathbf{x}_{1-\mathbf{m}}|\mathbf{m}, \mathbf{x_m})$ and $q(\mathbf{x}_{1-\mathbf{m}}|\mathbf{x_m})$. The KL divergence is non-negative and is minimized (to zero) if and only if the two distributions are equal:

$$p_{\mathbf{g}}(\mathbf{x}_{1-\mathbf{m}}|\mathbf{m},\mathbf{x_m}) = q(\mathbf{x}_{1-\mathbf{m}}|\mathbf{x_m}). \tag{63}$$

This demonstrates that the same optimality condition derived for the MCAR setting in Proposition 2 also holds under the MAR assumption.

## G  Experimental Setup

This appendix provides all information necessary for reproducibility: datasets, evaluation protocol, and computational resources. The hyperparameter settings and its sensitivity studies can be found in our supplemental material. Hyperparameters were selected through a sensitivity analysis. The synthetic datasets used in this sensitivity study consist of 1000 samples and 20 features. A missing rate of 20% was applied to each dataset. The data types include Gaussian, Uniform Correlated, and Mixed (Gaussian and Uniform). The results indicate that MIRI is *not sensitive* to the hyperparameters selections. See our supplementary material for more details.

### G.1  Datasets

**Synthetic Data.**   We generate $N = 6\,000$ samples in $\mathbb{R}^2$ by drawing two equally-sized clusters of $n = 3\,000$ points each from isotropic Gaussians. One cluster is centered at $(-2, -2)$ and the other at $(2, 2)$, both with standard deviation $\sigma = 0.5$.

**UCI Regression Benchmarks.**   Table 4 lists the ten UCI datasets used. For each, we report sample size and feature dimensionality.

Table 4: UCI datasets used in our study.

| Dataset | # Samples | # Features |
|---|---|---|
| wine | 1 599 | 11 |
| energy | 768 | 8 |
| parkinsons | 5 875 | 20 |
| stock | 536 | 11 |
| pumadyn32nm | 8 192 | 32 |
| housing | 506 | 13 |
| forest | 517 | 12 |
| bike | 17 379 | 17 |
| solar | 1 066 | 10 |
| gas | 2 565 | 128 |

**CIFAR-10.** We randomly sample 5 000 32×32 RGB images and apply pixel-level MCAR masks at varying rates. All three colour channels of each missing pixel are masked.

**CelebA.** We randomly sample 5 000 64×64 RGB images and apply channel-level MCAR masks at varying rates. All three colour channels of each missing pixel are masked independently.

## G.2 Evaluation Protocol

For each dataset, we generate ten independent MCAR masks and rerun the full training and imputation pipeline. Reported results are mean ± standard deviation over these runs.

## G.3 Hyperparameter Selection

Values were chosen via sensitivity analysis. Complete settings and search ranges are provided in our supplemental material. MIRI exhibits *low sensitivity* to these choices.

## G.4 Computational Resources

**Tabular Experiments.** NVIDIA P100 GPU (16 GB), Intel Xeon E5-2680 v4 CPU (8 cores, 2.4 GHz), 24 GB RAM.

**Image Experiments.** NVIDIA RTX 3090 GPU (24 GB), Intel Xeon Gold 6330 CPU (14 cores, 2.0 GHz), 90 GB RAM.

## G.5 Baseline Implementations

We evaluate **HyperImpute**, **MICE**, **MIWAE**, and **Sinkhorn** using the official HyperImpute repository with default settings.[*]

We adapt the official implementations of **KnewImp**[†], **TabCSDI**[‡], **GAIN**[§], and **TDM**[¶] from their respective GitHub repositories. **MissDiff** is reimplemented based on the algorithm described in the original publication [27]. All baselines use default hyperparameters unless otherwise stated.

# H Additional Experimental Results

## H.1 Performance on Synthetic Data

We evaluate each imputation method using three criteria, computed only on entries originally masked under the MCAR mechanism: (1) Root Mean Square Error (RMSE); (2) Mean Absolute Error (MAE); (3) Maximum Mean Discrepancy (MMD). Table 5 reports the mean ± standard deviation over ten independent MCAR masks (30% missing).

Overall, HyperImpute is optimal when minimizing per-entry error, achieving roughly 8% lower RMSE and 14% lower MAE than MIRI. In contrast, MIRI reduces MMD by more than 60% relative to HyperImpute, thereby better preserving the underlying data distribution despite a modest increase in point-wise error. MissDiff offers a balanced trade-off, whereas GAIN, TabCSDI, and KnewImp underperform on both fronts.

## H.2 Additional UCI Regression Experiments

We comprehensively evaluate imputation performance on ten UCI regression benchmarks under MCAR, MAR, and MNAR settings (Figures 5, 6, and 7). In some high-dimensional or high-missingness scenarios, certain baselines failed; for example, HyperImpute produced runtime errors

---

[*] https://github.com/vanderschaarlab/hyperimpute.
[†] https://github.com/JustusvLiebig/NewImp.
[‡] https://github.com/pfnet-research/TabCSDI.
[§] https://github.com/jsyoon0823/GAIN.
[¶] https://github.com/hezgit/TDM.

Table 5: **Performance metrics at 30% missingness.** Metrics computed over 10 runs; values denote mean $\pm$ standard deviation. The best values are highlighted in bold.

| Method | RMSE ($\downarrow$) | MAE ($\downarrow$) | MMD ($\downarrow$) |
|---|---|---|---|
| **GAIN** | $1.128 \pm 0.004$ | $0.600 \pm 0.002$ | $0.342 \pm 0.015$ |
| **TabCSDI** | $1.128 \pm 0.004$ | $0.600 \pm 0.002$ | $0.337 \pm 0.010$ |
| **KnewImp** | $1.130 \pm 0.004$ | $0.600 \pm 0.002$ | $0.335 \pm 0.009$ |
| **MissDiff** | $0.951 \pm 0.008$ | $0.437 \pm 0.004$ | $0.189 \pm 0.009$ |
| **HyperImpute** | $\mathbf{0.862 \pm 0.020}$ | $\mathbf{0.278 \pm 0.006}$ | $0.094 \pm 0.018$ |
| **MIRI (Ours*)** | $0.938 \pm 0.022$ | $0.325 \pm 0.009$ | $\mathbf{0.036 \pm 0.007}$ |

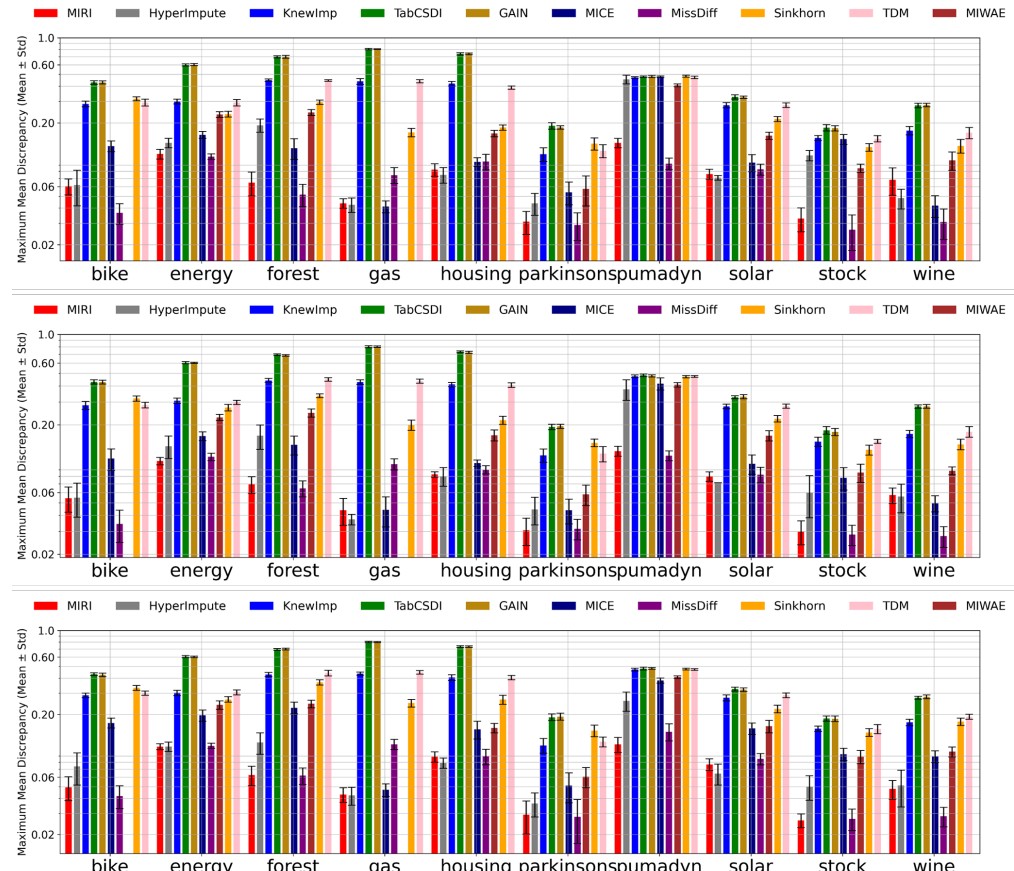

Figure 5: MCAR MMD on 10 UCI datasets (Above: 20% missingness, Middle: 40% missingness, Below: 60 % missingness). The lower the better.

and MIWAE encountered out-of-memory (OOM) errors. The results confirm that MIRI consistently delivers strong distributional fidelity, remains robust under high missing rates, and scales effectively to high-dimensional datasets.

### H.3 Additional CIFAR-10 Experiments

We evaluate the imputation quality of MIRI, GAIN, KnewImp, and HyperImpute on 15 randomly selected CIFAR-10 images corrupted under pixel-level MCAR with missing rates of 20%, 40% and 60% (see Figures 8(a), 8(b) and 8(c)). Across all missing-data regimes, MIRI consistently produces accurate, visually coherent reconstructions, in agreement with the quantitative metrics in Table 2. Although HyperImpute achieves competitive performance at 20% missingness, MIRI outperforms it at both 40% and 60% missingness.

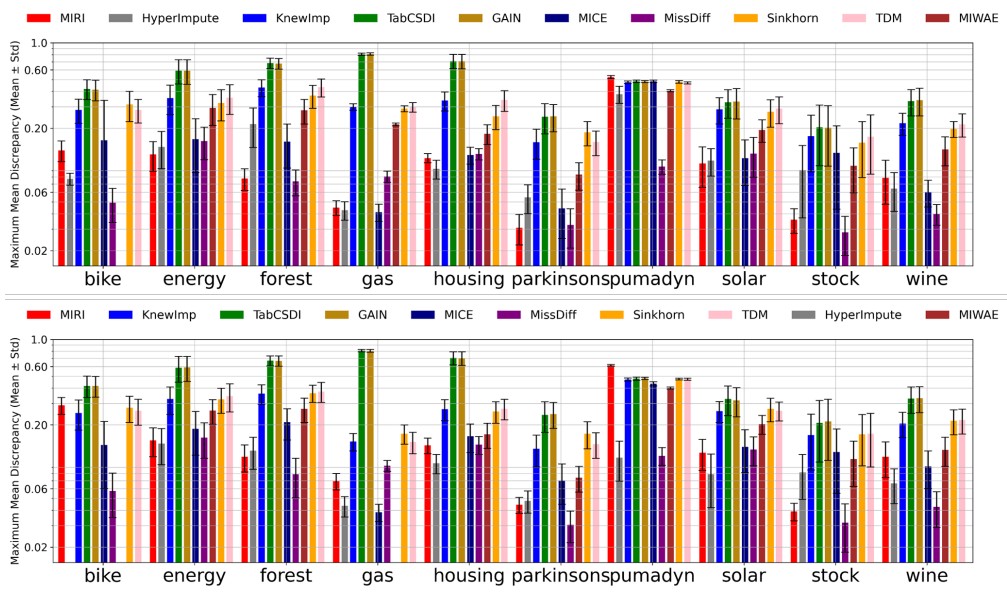

Figure 6: MAR MMD on 10 UCI datasets (Above: 40% missingness, Below: 80 % missingness). The lower the better.

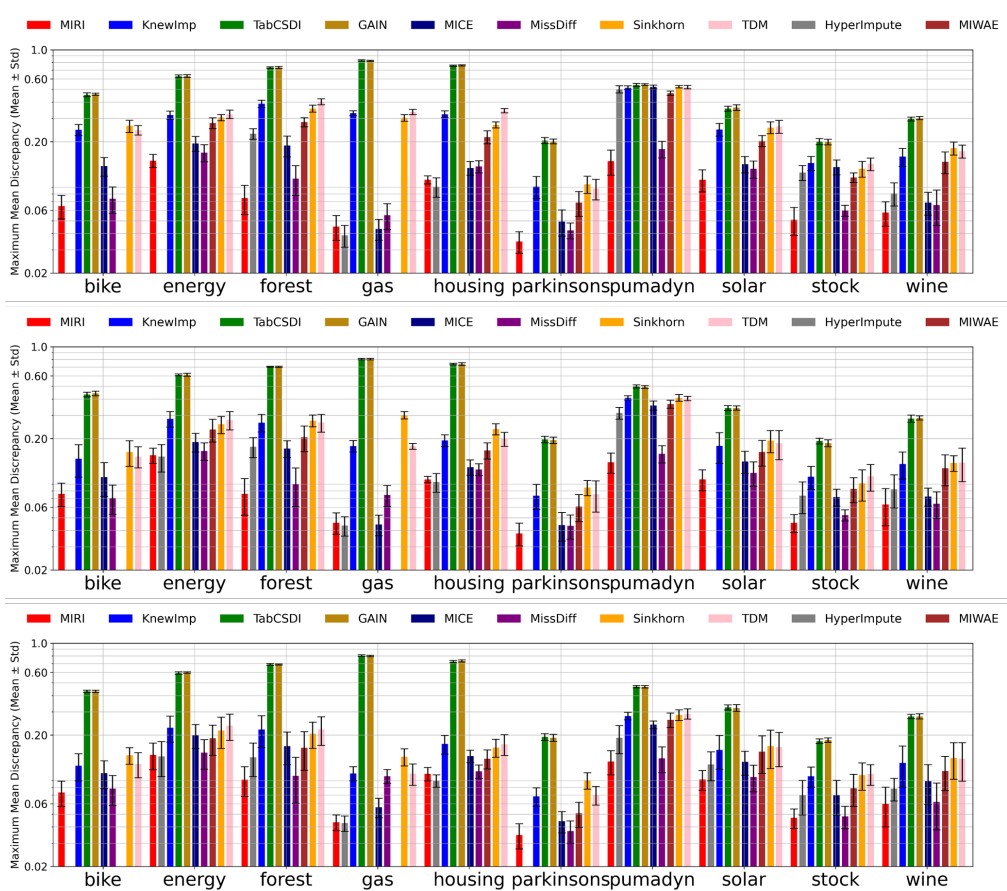

Figure 7: MNAR MMD on 10 UCI datasets (Above: 20% missingness, Middle: 40% missingness, Below: 60 % missingness). The lower the better.

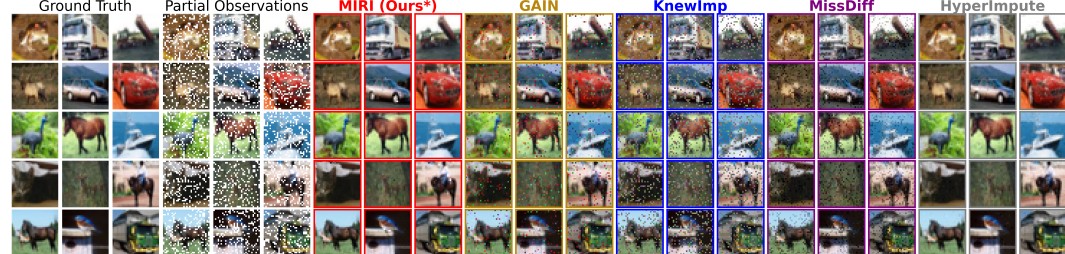

(a) 15 uncurated 32×32 CIFAR-10 images and their imputations. 20% of pixels randomly removed from *all RGB channels*.

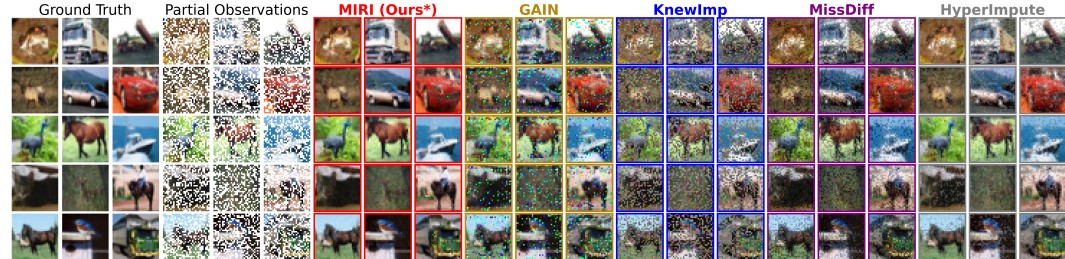

(b) 15 uncurated 32×32 CIFAR-10 images and their imputations. 40% of pixels randomly removed from *all RGB channels*.

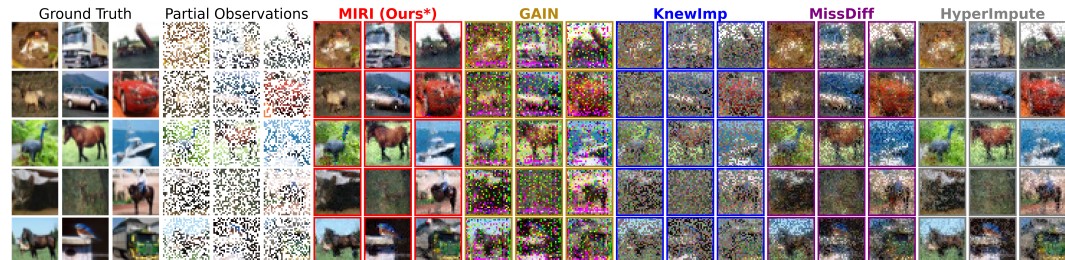

(c) 15 uncurated 32×32 CIFAR-10 images and their imputations. 60% of pixels randomly removed from *all RGB channels*.

# I Computational Time

Computational cost matters in high dimensions. MIRI is end-to-end vectorized ("vector in, vector out") and exploits parallel hardware common in image/tabular pipelines, whereas round-robin methods such as MICE and HyperImpute [12, 37] are inherently sequential and scale poorly with dimensionality.

**Wall-clock runtimes.** Table 6 reports mean ± std hours to impute 1000 samples across dimensionalities. We include diffusion-style baselines (DiffPuter [39], TabCSDI [42]), round-robin (HyperImpute), distribution-matching (MOT [26], TDM [41]) and VAE (MIWAE [24]). MIRI is on par with diffusion-based methods and substantially faster than round-robin in high dimensions. On CIFAR-10 (5k images, 60% missing), HyperImpute did not finish within a 24h budget on our PyTorch+CUDA setup, while MIRI finished in ≈3.5h on the same hardware.

# J Euler ODE Solver

In this section, we provide the Euler solver used in our experiments.

Table 6: Average computation time (hours) for 1000 samples across dimensions (mean±std).

| Method | 50d | 200d | 500d | 1000d | 2000d | 5000d |
|---|---|---|---|---|---|---|
| HyperImpute | $0.08 \pm 0.03$ | $0.23 \pm 0.02$ | $1.18 \pm 0.51$ | $3.84 \pm 0.99$ | $11.65 \pm 3.34$ | $57.54 \pm 0.00$ |
| TabCSDI | $0.18 \pm 0.00$ | $0.20 \pm 0.01$ | $0.26 \pm 0.00$ | $0.36 \pm 0.00$ | $0.53 \pm 0.01$ | $1.10 \pm 0.01$ |
| DiffPuter | $2.30 \pm 0.01$ | $3.22 \pm 0.18$ | $5.06 \pm 0.01$ | $7.70 \pm 0.05$ | $12.40 \pm 0.21$ | $29.10 \pm 0.00$ |
| MIWAE | $0.11 \pm 0.00$ | $0.30 \pm 0.00$ | $0.90 \pm 0.03$ | $2.02 \pm 0.01$ | OOM | OOM |
| MOT | $0.31 \pm 0.00$ | $0.33 \pm 0.00$ | $0.36 \pm 0.00$ | $0.38 \pm 0.00$ | $0.49 \pm 0.02$ | $0.56 \pm 0.00$ |
| TDM | $0.32 \pm 0.00$ | $0.55 \pm 0.00$ | $1.30 \pm 0.02$ | $3.30 \pm 0.04$ | $13.90 \pm 0.83$ | OOM |
| **MIRI (Ours*)** | $0.17 \pm 0.00$ | $0.21 \pm 0.00$ | $0.28 \pm 0.02$ | $0.37 \pm 0.02$ | $0.59 \pm 0.02$ | $1.30 \pm 0.01$ |

---

**Algorithm 2** ODE Solver with Euler Method

---

**Require:** Velocity field $\mathbf{v}$, Initial condition $\{(\mathbf{X}_i, \mathbf{M}_i)\}$, Number of Euler steps $N$.

1: **for** $k = 1$ to $N$ **do**

2: $\quad \tau = \dfrac{k}{N}$

3: $\quad \forall i, \mathbf{X}_i \leftarrow \mathbf{X}_i + \dfrac{1}{N} \cdot \mathbf{v}(\mathbf{X}_i, \mathbf{M}_i, \tau)$

4: **end for**

5: **return** $\{\mathbf{X}_i\}$

---

