# OpenReview forum: "Missing Data Imputation by Reducing Mutual Information with Rectified Flows"
_NeurIPS.cc/2025/Conference — NeurIPS 2025 poster_

### Official Review · Reviewer_k1XP · 2025-06-29

**Clarity:** 3
**Significance:** 2
**Originality:** 3
**Rating:** 4
**Confidence:** 4

**Summary:**

The authors introduced a missing data imputation method that follows along findings around GAIN's strengths, i.e., by enforcing the orthogonality between missing data and data missingness patterns. Specifically, the authors introduces MIRI that is built on top of rectified flows where the objectives can be formulated as ODEs. Experiments were ran on three data set with missing-completely-at-random (MCAR) missingness setup.

**Questions:**

Following from the weaknesses above, the reviewer would appreciate additional clarifications around, i.e.,

* Would MIRI require data to be MCAR? If so, what would make a difference when imputing MAR data and how does MIRI perform on MAR datasets (e.g., MIMIC, physionet or synthetic MAR masking).
* Would MIRI also work for imputing time-series? If so, the reviewer is curious in how it compares to other diffusion-based methods in time-series imputation.
* How would MIRI benefit downstream classification/regression tasks using the imputed data? Could authors quantify and show the improvements in experiments?

**Ethical Concerns:**

["NO or VERY MINOR ethics concerns only"]

**Final Justification:**

Most concerns were resolved during rebuttal so moved my recommendation to the acceptance side.

**Limitations:**

The authors acknowledged limitations in computing resources and for part of their theories. However, to the reviewer's point, the more important discussion is missing, i.e., if the approach can work with MAR data and for time-series (see the weaknesses section above).

**Paper Formatting Concerns:**

The conclusion section is missing.

**Quality:**

2

**Strengths And Weaknesses:**

## Strengths

* Motivation and methodology are clearly written.
* Theoretical analyses are provided.

## Weaknesses

* Unclarity of the problem setup and the actual use-cases/assumptions of MIRI. The approach sits on the basis that imputed data can be orthogonal to the missingness patterns, which is basically the MCAR setting. However, it was not mentioned specifically if MIRI can only work work MCAR or it can be extended to MAR which covers many real-world use cases in healthcare data and other time-series extensively. Also looks like all the experiments are ran based on the MCAR setup.
* Moreover, imputation for time-series [see 1-14 below as a non-exhaustive list] is an important aspect of missing data imputation and has been extensively studied. While it does not seem the author claimed if MIRI would work for time-series imputation, the experiments used datasets that are not time-seires.
* Missing discussion and comparison to related works. Recently there are a few works that emphasized if perfect imputation are needed for  downstream tasks [13-16], i.e., [15, 16] argued that mean imputation could be sufficient for downstream tasks, [13] showed that missingness patterns themselves could provide useful information for downstream tasks, with both [13] and [14] introduced methods that can directly achieve downstream tasks without the need for imputation.
* Missing experimental setup. A majority of existing imputation approaches (i.e., listed above) also focused on how their imputed data could benefit the performance of downstream classification/regression tasks, as part of experiments. Such experiments are missing in this paper.
* Missing baselines and related work in using diffusion models for missing data imputation. Upon some preliminary search there also exists a number of papers that use diffusion to impute missing data [4-7]. It would be helpful to discuss them in the paper and compare with them in experiments if possible.

[1] Luo, Yuan, Peter Szolovits, Anand S. Dighe, and Jason M. Baron. "3D-MICE: integration of cross-sectional and longitudinal imputation for multi-analyte longitudinal clinical data." Journal of the American Medical Informatics Association 25, no. 6 (2018): 645-653.

[2] Fortuin, Vincent, Dmitry Baranchuk, Gunnar Rätsch, and Stephan Mandt. "Gp-vae: Deep probabilistic time series imputation." In International conference on artificial intelligence and statistics, pp. 1651-1661. PMLR, 2020.

[3] Xinyu Yang, Yu Sun, Xiaojie Yuan, and Xinyang Chen. Frequency-aware generative models for multivariate time series imputation. In The Thirty-eighth Annual Conference on Neural Information Processing Systems, 2024

[4] Lopez Alcaraz, Juan Miguel, and Nils Strodthoff. "Diffusion-based time series imputation and forecasting with structured atate apace models." Transactions on machine learning research (2023): 1-36.

[5] Jianping Zhou, Junhao Li, Guanjie Zheng, Xinbing Wang, and Chenghu Zhou. Mtsci: A conditional diffusion model for multivariate time series consistent imputation. In CIKM, pages 3474–3483, 2024.

[6] Xu Wang, Hongbo Zhang, Pengkun Wang, Yudong Zhang, Binwu Wang, Zhengyang Zhou, and Yang Wang. An observed value consistent diffusion model for imputing missing values in multivariate time series. In SIGKDD, 2023.

[7] Yusuke Tashiro, Jiaming Song, Yang Song, and Stefano Ermon. CSDI: Conditional score-based diffusion models for probabilistic time series imputation. In NeurIPS, 2021.

[8] Shuai Liu, Xiucheng Li, Gao Cong, Yile Chen, and Yue Jiang. Multivariate time-series imputation with disentangled temporal representations. 2022.

[9] Yonghong Luo, Ying Zhang, Xiangrui Cai, and Xiaojie Yuan. E2GAN: End-to-end generative adversarial network for multivariate time series imputation. In IJCAI, 2019.

[10] Guojun Liang, Prayag Tiwari, Sławomir Nowaczyk, and Stefan Byttner. Higher-order spatio-temporal physics-incorporated graph neural network for multivariate time series imputation. In CIKM, 2024.

[11] Andrea Cini, Ivan Marisca, and Cesare Alippi. Filling the g ap s: Multivariate time series imputation by graph neural networks. In ICLR, 2022.

[12] Zhengping Che, Sanjay Purushotham, Kyunghyun Cho, David Sontag, and Yan Liu. Recurrent neural networks for multivariate time series with missing values. Scientific Reports, 8(1), Apr 2018.

[13] Gao, Qitong, Dong Wang, Joshua David Amason, Siyang Yuan, Chenyang Tao, Ricardo Henao, Majda Hadziahmetovic, Lawrence Carin, and Miroslav Pajic. "Gradient Importance Learning for Incomplete Observations." In International Conference on Learning Representations.

[14] Cao, Wei, Dong Wang, Jian Li, Hao Zhou, Lei Li, and Yitan Li. "Brits: Bidirectional recurrent imputation for time series." Advances in neural information processing systems 31 (2018).

---

> ### Author Rebuttal · Authors · 2025-07-30
>
> # MAR, MNAR
> > Would MIRI require data to be MCAR? If so, what would make a difference when imputing MAR data and how does MIRI perform on MAR datasets ...
>
> Our method works on **MAR and MCAR** data and we provide both theoretical and empirical justification.
>
> In the MAR setting, the independent assumption becomes $X_{\mathrm{miss}} \perp M | X_{\mathrm{obs}}$. Following the same rationale of MIRI, we can minimize the **conditional mutual information** given the observed variables.
> We can show the condition in Proposition 2 is also the sufficient condition to minimize the conditional mutual information. In other words, running the unmodified MIRI on MAR data will reduce the conditional mutual information $\mathrm{I}[X_{\mathrm{miss}}(g); M | X_{\mathrm{obs}}]$, justifying MIRI's capability on MAR datasets.
>
> We provide a proof at the end of this reply.
>
> We also conduct a series of experiments to evaluate MIRI's performance under **MAR and MNAR** settings, following settings in [1]. The results demonstrate the robustness and effectiveness of our method.
>
> At 40\% missingness, MIRI demonstrates highly competitive performance. As shown in the table below, it achieves the best (lowest) MMD on three over six datasets, (gas, parkinsons, solar) and is a close second on the rest.
>
> **MMD of MAR imputations on UCI datasets ($p_{\mathrm{miss}}=0.4$). 50% of observed variables are used to generate mask.**
> |**Method**|**forest**|**gas**|**housing**|**parkinsons**|**solar**|**stock**|
> |-|-|-|-|-|-|-|
> |HyperImpute|0.243 ± 0.059|0.049 ± 0.008|**0.107 ± 0.021**|0.067 ± 0.006|0.109 ± 0.028|0.127 ± 0.018|
> |GAIN|0.628 ± 0.038|0.807 ± 0.014|0.624 ± 0.082|0.316 ± 0.057|0.313 ± 0.086|0.249 ± 0.071|
> |TabCSDI|0.641 ± 0.046|0.801 ± 0.019|0.626 ± 0.064|0.327 ± 0.065|0.311 ± 0.081|0.248 ± 0.076|
> |MissDiff|**0.072 ± 0.008**|0.081 ± 0.011|0.124 ± 0.007|0.042 ± 0.016|0.098 ± 0.029|**0.028 ± 0.005**|
> |MOT|0.408 ± 0.045|0.302 ± 0.003|0.181 ± 0.014|0.228 ± 0.036|0.267 ± 0.082|0.178 ± 0.059|
> |MIWAE|0.313 ± 0.030|0.216 ± 0.006|0.136 ± 0.014|0.091 ± 0.014|0.166 ± 0.048|0.108 ± 0.030|
> |TDM [1]|0.471 ± 0.028|0.301 ± 0.015|0.249 ± 0.027|0.188 ± 0.052|0.270 ± 0.059|0.201 ± 0.065|
> |**MIRI (Ours\*)**|0.084 ± 0.020|**0.045 ± 0.008** |0.121 ± 0.003|**0.037 ± 0.012**|**0.092 ± 0.021**|0.037 ± 0.007|
>
> MIRI also demonstrates highly competitive performance on MNAR data. It achieves the best (lowest) MMD on five over six datasets and is a close second on the rest.
>
> **MMD of MNAR imputations on UCI datasets ($p_{\mathrm{miss}}=0.4$). All variables are used to generate mask.**
> |**Method**|**forest**|**gas**|**housing**|**parkinsons**|**solar**|**stock**|
> |-|-|-|-|-|-|-|
> |HyperImpute|0.178 ± 0.008|0.049 ± 0.005|**0.087 ± 0.017**|Error|Error|0.069 ± 0.010|
> |GAIN|0.704 ± 0.005|0.805 ± 0.002|0.750 ± 0.006|0.197 ± 0.011|0.325 ± 0.004|0.179 ± 0.008|
> |TabCSDI|0.704 ± 0.006|0.810 ± 0.013|0.739 ± 0.010|0.195 ± 0.013|0.327 ± 0.006|0.190 ± 0.004|
> |MissDiff|0.085 ± 0.028|0.076 ± 0.015|0.116 ± 0.007|0.049 ± 0.005|0.093 ± 0.014|0.057 ± 0.004|
> |MOT|0.259 ± 0.036|0.307 ± 0.019|0.262 ± 0.017|0.089 ± 0.018|0.179 ± 0.027|0.098 ± 0.005|
> |MIWAE|0.220 ± 0.042|OOM|0.186 ± 0.012|0.071 ± 0.017|0.146 ± 0.027|0.087 ± 0.002|
> |TDM [1]|0.284 ± 0.056|0.176 ± 0.006|0.209 ± 0.007|0.098 ± 0.020|0.171 ± 0.042|0.111 ± 0.001|
> |**MIRI (Ours\*)**|**0.066 ± 0.014**|**0.045 ± 0.008**|0.097 ± 0.002|**0.043 ± 0.007**|**0.093 ± 0.003**|**0.052 ± 0.002**|
>
> [1] Zhao, He, et al. "Transformed distribution matching for missing value imputation." International Conference on Machine Learning. PMLR, 2023.
>
> We will add these theoretical and empirical results in the revision.
>
> ---
>
> # Time-series
> > Would MIRI also work for imputing time-series? If so, the reviewer is curious in how it compares to other diffusion-based methods in time-series imputation.
>
> Yes, MIRI works on time-series. We can consider a **self-supervised version** of MIRI.
>
> 1. We delete all missing values from the time-series and obtain an zero-padded dataset $X_0$.
> 2. Similar to CSDI [2], we randomly sample many masks to form pairs of samples $(X_0, M)$ and run MIRI on it.
> 3. We impute the original time-series using vector fields obtained from MIRI.
>
> This approach is similar to the self-supervised scheme used by CSDI: delete missing values first, if the time-series is not missing too many values, the imputer trained on the remaining time-series should also be a good imputer for the original time-series.
>
> Comparing to the existing diffusion-based method (CSDI), our method can impute future time-series with unknown missing lengths while **the imputation space of CSDI is rigid**. CSDI learns to sample from $\mathbb{P}(X_\mathrm{miss} | X_\mathrm{obs})$ using a conditional diffusion model. Once a diffusion model is trained, the dimension of $X_\mathrm{miss}$ is fixed. Thus, the diffusion model trained on one time-series cannot be used to impute other time-series if the lengths/sizes of the missing patterns are different. However, in our algorithm, the algorithm is agnostic to the space of $X_\mathrm{miss}$: MIRI does not learn $\mathbb{P}(X_\mathrm{miss} | X_\mathrm{obs})$ but only to finds $X_\mathrm{imp}$ that satisfies some independence condition. MIRI vector fields trained on one time-series can be easily used to impute other time-series with different missing lengths/sizes.
>
> Due to this flexibility, we can consider a large foundation model, trained by MIRI on many fully observed or partially observed sequences, then use the pretrained vector field to impute/predict any similar unseen time-series.
>
> We have run MIRI using this algorithm on real-world time-series and obtain promising results as in the below table. For the baselines, we adopted the PyPOTS [5] library with fair experimental settings. The datasets were loaded from the statsmodels library.
>
> **RMSE of imputations on time‐series datasets (MCAR, $p_\mathrm{mis}=0.3$).**
> |Method|Airpassengers|CO2|Nile|Sunspots|Toy (Sine)|
> |---|---|---|---|---|---|
> |CSDI [2]|21.658±13.127|13.954±14.451|33.806±30.647|23.128±7.211|7.105±4.396|
> |TimeMixer++ [3] |0.989±0.070|0.944±0.024|**1.056±0.109**|0.964±0.115|0.958±0.036|
> |TSLANet [4] |1.010±0.062|0.956±0.045|1.060±0.106|0.966±0.084|0.949±0.050|
> |**MIRI (Ours\*)**|**0.310±0.050**|**0.342±0.163**|1.107±0.159|**0.518±0.117**|**0.460±0.106**|
>
> We will discuss this extension to time-series in the revision of our paper and include these empirical results.
>
> [2] Tashiro et al., CSDI: Conditional Score-based Diffusion Models for Probabilistic Time Series Imputation. NeurIPS 2021.
>
> [3] Wang et al., TimeMixer++: A General Time Series Pattern Machine for Universal Predictive Analysis. ICLR 2025.
>
> [4] Eldele et al., TSLANet: Rethinking Transformers for Time Series Representation Learning. ICML 2024.
>
> [5] Du. PyPOTS: a Python toolbox for data mining on Partially-Observed Time Series, 2023.
>
> ---
>
> # Downstream Performance
> > How would MIRI benefit downstream classification/regression tasks using the imputed data? Could authors quantify and show the improvements in experiments?
>
> We thank the reviewer for raising this issue. We have performed a 10-class classification experiment on imputed CIFAR-10 imputations and the results are shown below.
>
> | **Method**         | **20 %** | **40 %** | **60 %** |
> |--------------------|--------------:|--------------:|--------------:|
> | MissDiff           | 0.486        | 0.269        | 0.152        |
> | KnewImp            | 0.267        | 0.137        | 0.106        |
> | GAIN               | 0.337        | 0.133        | 0.096        |
> | HyperImpute        | 0.804        | 0.405        | 0.212        |
> | **MIRI (Ours\*)**  | **0.812**    | **0.525**    | **0.364**    |
>
> It can be seen that the proposed method achieves the highest overall accuracy among all compared imputation methods and we will add this result to the revision.
>
> ---
>
> # Proof for MAR Setting
>
> Define the notation:  $X_\mathrm{obs} = X_M$ and $X_\mathrm{miss} = X_{1-M}$, and consider the goal of minimizing the **conditional mutual information**:
>
> $$\mathrm{I}[X_\mathrm{miss}(g); M | X_\mathrm{obs}] = E_{(x,m)\sim (X,M)}\log \frac{p_g(x_{1-m}, m |x_m )}{p(x_{1-m}| x_m)p(m|x_m)}$$
>
> Following the a very similar proof in proposition 1, we have the iterative algorithms to reduce the mutual information. At iteration $t$:
>
> - $$
>   g^* = \arg\min_{g} E_{(x,m) \sim (X,M)}\left[\log \frac{p(x_{1-m}, m \mid x_m )}{q(x_{1-m} \mid x_m) q(m \mid x_m)}\right],
>   $$
>   where $p$ is the density function of $P_{X^{(t)}, M}$ and $q$ is the density function of $P_{X^{(t-1)}}$, the probability of $X$ in the previous iteration.
>
> - Impute using $g^*$ and repeat.
>
> We now show that  $p(x_{1-m} \mid m, x_m) = q(x_{1-m} \mid x_m)$  is a sufficient condition for $g$ to be optimal in the first step.
>
> Analogous to Proposition 2, we can rewrite the objective function:
>
> \\[
> \begin{aligned}
> E_{(x,m) \sim (X,M)}\left[\log \frac{p(x_{1-m}, m \mid x_m )}{q(x_{1-m} \mid x_m)\, p(m \mid x_m)}\right] =&E_{(x,m) \sim (X,M)}\left[\log \frac{p(x_{1-m} \mid m, x_m )}{q(x_{1-m} \mid x_m)}\right] \\\\
> =&E_{(m, x_m) \sim (M, X_M)} E_{x_{1-m} \sim X_{1-M} \mid M, X_M} \left[\log \frac{p(x_{1-m} \mid m, x_m )}{q(x_{1-m} \mid x_m)}\right]
> \end{aligned}
> \\]
>
> Note that $p(m \mid x_m)$ does not change across iterations, so $p(m \mid x_m) = q(m \mid x_m)$.
> The inner expectation is a KL divergence between $p(x_{1-m} \mid m, x_m)$ and $q(x_{1-m} \mid x_m)$, which is minimized (to zero) when the two distributions are equal:
>
> \\[
> p(x_{1-m} \mid m, x_m) = q(x_{1-m} \mid x_m)
> \\]
>
> This is exactly the sufficient condition stated in Proposition 2.

---

> > ### Comment · Reviewer_k1XP · 2025-08-04
> >
> > The reviewer would like to thank the authors for the detailed responses. Most of the reviewer's concerns are addressed and the evaluation of the paper will be updated accordingly.

---

> > > ### Author Response · Authors · 2025-08-04
> > > **Thank you!**
> > >
> > > We thank the reviewer for insightful comments and time and effort for reviewing our paper!
> > >
> > > It is great to hear that most concerns are addressed!
> > >
> > > If any further question arises, please do not hesitate to comment and we are happy to respond during the discussion period.

---

### Official Review · Reviewer_yRf4 · 2025-06-30

**Clarity:** 3
**Significance:** 3
**Originality:** 3
**Rating:** 4
**Confidence:** 3

**Summary:**

The authors develop multivariate imputation engines, constructed using a so-called imputation ODE obtained through Rectified Flow training. The author's imputation engine is optimal due to the adversarial nature of their training: they optimize their imputation engines to minimize the KL divergence between the joint distribution of the missingness masks and study variables and a product of the marginal distributions of two. Iterative estimation algorithms are introduced which successively decrease the mutual information between X and M, so that that it becomes difficult to distinguish imputed from observed values.  The authors also connect their approach (MIRI) to popular iterative imputation methods like MICE. The effectiveness of their methods are demonstrated in two simulation studies.

**Questions:**

Can the authors think more about how their method situates among MAR vs. MNAR imputation methods?

**Ethical Concerns:**

["NO or VERY MINOR ethics concerns only"]

**Final Justification:**

I think the authors have addressed the limitations of this work, namely that it is restricted to MAR problems in their responses, edits to the paper, and acknowledgments that this is a future area for research. As such, I recommend the paper for a borderline accept.

**Limitations:**

The authors should be more transparent about the fact that their methods are specifically designed for the case where Y_{j}^{mis} (The missing portion of Y_j) is conditionally independent of M given Y^{obs} (observed data). In this setting we want to preserve the distribution of the observed data among the imputed values, which the proposed methods are optimized to do. Unless, I am missing something, it seems that the methods presented are almost tailored to the MAR, and not MNAR case.

**Quality:**

3

**Strengths And Weaknesses:**

I enjoyed this fresh perspective on missing data imputation. The idea to treat imputation as an adversarial learning problem is quite interesting. The author's formulation is quite interesting -- to build a generative imputation engine that is optimized such that joint imputations are essentially indistinguishable from the observed data based on their mutual information with the missingness mask.

Weakness: This method is restricted to MAR problems. With informative missingness (MNAR). The missing data is dependent on the missingness process, so this formulation would not work. The authors should be sure to clarify this limitation in their manuscript more clearly.

---

> ### Author Rebuttal · Authors · 2025-07-30
>
> # MAR, MNAR
>
> > This method is restricted to MAR problems... Can the authors think more about how their method situates among MAR vs. MNAR imputation methods?
>
> Thanks for pointing it out! We will make sure to include a paragraph in the introduction to clarify missingness assumption of the proposed method works.
>
> Our method is motivated by the key assumption of MCAR, that true data is independent of the mask. For MCAR data, the proposed method reduces the mutual information. For MAR data, the proposed method reduces the conditional mutual information, given the observed variables (see proof at the end of the reply).
>
> However, for MNAR dataset, the ground truth could be _highly correlated_ with the missing mask, thus minimizing any type of mutual information is unlikely to work. To impute MNAR datasets, one must have some knowledge about the missingness pattern and _learn the probability of missing_. Our method does not incorporate such an assumption, thus will produce a biased result. However, extending MIRI to MNAR dataset is certainly an interesting future direction.
>
> It is also worth noting that, we can view other imputation algorithms as MIRI under various modeling and optimization approximations (Section 5), meaning these popular imputation methods are also unlikely to work under the MNAR setting as they are also implicitly minimizing the mutual information between $X$ and $M$. We will include these discussions in the revision.
>
> We include some additional empirical results on MNAR data below.
>
> >The authors should be more transparent about the fact that their methods are specifically designed for the case where $Y_{j}^{mis}$ (The missing portion of $Y_j$) is conditionally independent of $M$ given $Y^{obs}$ (observed data).
>
> Thank you for raising this point. We will definitely clarify our assumption in the revision!
>
> ---
>
> # MNAR Experiments
>
> As empirical validation, we also performed additional experiments under MNAR settings. As shown in the table below, although our proposed method is _not_ designed for MNAR, it still consistently delivers low MMD across the baselines. Note that we also added a few other baselines that other reviewers requested and please see these reviews for references.
>
> **MMD of MNAR imputations on UCI datasets ($p_{\mathrm{mis}}=0.4$). All variables are used to generate mask.**
> | **Method**        | **forest**          | **gas**             | **housing**          | **parkinsons**       | **solar**           | **stock**           |
> |-------------------|---------------------|---------------------|----------------------|----------------------|---------------------|---------------------|
> | HyperImpute       | 0.178 ± 0.008       | 0.049 ± 0.005       | **0.087 ± 0.017**    | Error                | Error               | 0.069 ± 0.010       |
> | GAIN              | 0.704 ± 0.005       | 0.805 ± 0.002       | 0.750 ± 0.006        | 0.197 ± 0.011        | 0.325 ± 0.004       | 0.179 ± 0.008       |
> | TabCSDI           | 0.704 ± 0.006       | 0.810 ± 0.013       | 0.739 ± 0.010        | 0.195 ± 0.013        | 0.327 ± 0.006       | 0.190 ± 0.004       |
> | MissDiff          | 0.085 ± 0.028       | 0.076 ± 0.015       | 0.116 ± 0.007        | 0.049 ± 0.005        | 0.093 ± 0.014       | 0.057 ± 0.004       |
> | MOT          | 0.259 ± 0.036       | 0.307 ± 0.019       | 0.262 ± 0.017        | 0.089 ± 0.018        | 0.179 ± 0.027       | 0.098 ± 0.005       |
> | MIWAE             | 0.220 ± 0.042       | OOM                 | 0.186 ± 0.012        | 0.071 ± 0.017        | 0.146 ± 0.027       | 0.087 ± 0.002       |
> | TDM               | 0.284 ± 0.056       | 0.176 ± 0.006       | 0.209 ± 0.007        | 0.098 ± 0.020        | 0.171 ± 0.042       | 0.111 ± 0.001       |
> | **MIRI (Ours\*)** | **0.066 ± 0.014**   | **0.045 ± 0.008**   | 0.097 ± 0.002        | **0.043 ± 0.007**    | **0.093 ± 0.003**   | **0.052 ± 0.002**   |
>
> We will add these additional empirical results in the revision.
>
> ---
>
> # Proof for MAR Setting
>
> Define the notation:  $X_\mathrm{obs} = X_M$ and $X_\mathrm{miss} = X_{1-M}$, and consider the goal of minimizing the **conditional mutual information**:
>
> $$\mathrm{I}[X_\mathrm{miss}(g); M | X_\mathrm{obs}] = E_{(x,m)\sim (X,M)}\log \frac{p_g(x_{1-m}, m |x_m )}{p(x_{1-m}| x_m)p(m|x_m)}$$
>
> Following the a very similar proof in proposition 1, we have the iterative algorithms to reduce the mutual information. At iteration $t$:
>
> - $$
>   g^* = \arg\min_{g} E_{(x,m) \sim (X,M)}\left[\log \frac{p(x_{1-m}, m \mid x_m )}{q(x_{1-m} \mid x_m) q(m \mid x_m)}\right],
>   $$
>   where $p$ is the density function of $P_{X^{(t)}, M}$ and $q$ is the density function of $P_{X^{(t-1)}}$, the probability of $X$ in the previous iteration.
>
> - Impute using $g^*$ and repeat.
>
> We now show that  $p(x_{1-m} \mid m, x_m) = q(x_{1-m} \mid x_m)$  is a sufficient condition for $g$ to be optimal in the first step.
>
> Analogous to Proposition 2, we can rewrite the objective function:
>
> \\[
> \begin{aligned}
> E_{(x,m) \sim (X,M)}\left[\log \frac{p(x_{1-m}, m \mid x_m )}{q(x_{1-m} \mid x_m)\, p(m \mid x_m)}\right] =&E_{(x,m) \sim (X,M)}\left[\log \frac{p(x_{1-m} \mid m, x_m )}{q(x_{1-m} \mid x_m)}\right] \\\\
> =&E_{(m, x_m) \sim (M, X_M)} E_{x_{1-m} \sim X_{1-M} \mid M, X_M} \left[\log \frac{p(x_{1-m} \mid m, x_m )}{q(x_{1-m} \mid x_m)}\right]
> \end{aligned}
> \\]
>
> Note that $p(m \mid x_m)$ does not change across iterations, so $p(m \mid x_m) = q(m \mid x_m)$.
> The inner expectation is a KL divergence between $p(x_{1-m} \mid m, x_m)$ and $q(x_{1-m} \mid x_m)$, which is minimized (to zero) when the two distributions are equal:
>
> \\[
> p(x_{1-m} \mid m, x_m) = q(x_{1-m} \mid x_m)
> \\]
>
> This is exactly the sufficient condition stated in Proposition 2.

---

> > ### Comment · Reviewer_yRf4 · 2025-08-05
> >
> > Thank you for this explanation. I am generally satisfied by the responses.

---

> ### Author Response · Authors · 2025-08-08
> **Thank you and Further Thoughts on MNAR Data**
>
> Dear Reviewer,
>
> Thank you again for your inspiring comments on our paper. We will incorporate our rebuttal into the revised manuscript and clearly state the assumptions underlying our method (**MCAR** and **MAR**, but not **MNAR**).
>
> ---
> Your feedback has also encouraged us to explore extending MIRI algorithm from handling MAR data to MNAR data.
>
> We can model the missing mask-generation process with the graphical structure  $ M \\leftarrow Z \\leftarrow X$, where $Z$ is a latent variable that depends on the ground-truth $X$. If $Z$ is a deterministic function of $X_{\\text{obs}}$, we recover the MAR setting. More generally, $Z$ can be viewed as a latent representation of $X$.
>
> If $Z$ were observable, we could simply define $V = (Z, X_{\\text{obs}})$, reducing the problem to MAR imputation with $V$ as the observed variables, yielding the conditional independence  $X_{\\text{miss}}{(g)} \\;\\bot\\!\\!\\!\\bot\\; M \\mid V$, which the MIRI method could work with.
>
> Because $Z$ is unobserved, we need to learn it through an auxiliary procedure. Below is a prototype algorithm that jointly learns $Z$ and imputes $X_{\\text{miss}}$:
>
> 1. **Randomly initialize $Z$.**
> 2.  Treating $V = (Z, X_{\\text{obs}})$ as observed, and **impute $X_{\\text{miss}}$** using MIRI,
> 3. **Train a model for $P(Z \\mid X, M)$** on the imputed data and sample a new $Z$.
> 4. **Repeat** steps 2–3 until convergence.
>
> This algorithm is also similar to the other MNAR algorithm GINA (Ma and Zhang, 2021), where authors jointly optimize an imputer $P(X_\mathrm{miss}| X_\mathrm{obs})$ as well as a latent variable model $P(Z|X_\mathrm{obs})$. The effectiveness of this approach hinges on modelling and training strategy for the variable $Z$. Computation may also be heavy, as MIRI now needs to be run multiple times.
>
> Nonetheless, we think this is a promising future direction. Investigating how well MIRI can operate in tandem with a latent-representation learning algorithm under an MNAR setting is an intriguing future work, and we will outline it in our revision.
>
> Ma and Zhang, Identifiable Generative Models for Missing Not at Random Data Imputation, NeurIPS2021
>
> ---
>
> Authors again thank the reviewer for inspiring feedback on our paper!

---

### Official Review · Reviewer_Dwu5 · 2025-07-01

**Clarity:** 2
**Significance:** 3
**Originality:** 3
**Rating:** 5
**Confidence:** 3

**Summary:**

This paper introduces MIRI, a novel method for missing data imputation based on rectified flows. The key idea is to iteratively reduce the mutual information between the imputed data and the missingness mask, which corresponds to minimizing the KL divergence between their joint distribution and the product of their marginals. The authors provide theoretical justification for this approach and show that several existing methods can be cast within this general framework. They present a practical iterative algorithm based on rectified flows to implement this objective and validate its performance across a variety of datasets. Extensive experiments show that MIRI consistently achieves strong results. The paper also discusses MIRI’s computational limitations in an informative manner.

**Questions:**

- The proposed method seems closely related to DiffPuter, which also applies iterative imputation via a generative model using an EM-like approach. In particular, the EM updates in Diffuser resemble SGD and the imputation steps of Algorithm 1. Could the authors elaborate further on the similarities and differences beyond the brief discussion in lines L258–262? A more in-depth comparison could clarify the novelty of MIRI and affect the originality rating.

- GAIN performs surprisingly poorly in the CIFAR MCAR experiments. Could the authors provide insight into this result? Was the model properly tuned or trained under comparable conditions? If GAIN is underperforming due to experimental choices, clarifying this could impact the validity of the empirical comparisons.

**Ethical Concerns:**

["NO or VERY MINOR ethics concerns only"]

**Final Justification:**

The authors have provided a detailed and thoughtful rebuttal that effectively addresses the main concerns raised in my initial review. I particularly appreciate the expanded comparison with DiffPuter, the discussion on the limitations under MNAR, and the inclusion of MIWAE as a strong baseline, which enhances both the theoretical positioning and the empirical rigor of the paper. While the clarity and motivation in the Introduction could still benefit from revision, I trust the authors will incorporate the reviewers’ suggestions in the final version. Overall, I find the contribution original and well-justified, and I have updated my score to reflect my strengthened confidence in the work.

**Limitations:**

The authors acknowledge the computational limitations of their approach, which is appreciated. However, a more explicit discussion is needed regarding the assumptions behind their method—particularly the reliance on MCAR missingness mechanisms. MIRI is unlikely to perform well in MNAR settings, which are common in real-world data, and this should be clearly stated in the paper. Including a short discussion of this limitation in the introduction or a dedicated paragraph in Section 7 would help set clearer expectations about the scope of the method.

**Paper Formatting Concerns:**

I didn't notice any formatting issues.

**Quality:**

3

**Strengths And Weaknesses:**

## Strengths
- The paper proposes a novel and theoretically grounded approach to missing data imputation using rectified flows. The method provides a clear and principled objective that unifies and generalizes several prior works.
- Theoretical results are solid and insightful. The authors rigorously show how existing approaches can be reinterpreted within the MIRI framework.
- Empirical results across a variety of datasets support the effectiveness of the proposed method. The paper also transparently discusses MIRI’s computational limitations, which adds to its credibility.

## Weaknesses
### Main concerns
- The motivation and presentation in the introduction could be clearer. The writing flow is somewhat disjointed, making it difficult to understand what specific challenges MIRI addresses and how it stands out from prior work. Important limitations of existing approaches (e.g., GAIN) that are only discussed later should be summarized earlier in the paper.
- The literature review underrepresents recent progress in deep generative modelling approaches for missing data imputation, particularly VAE-based imputation methods (e.g. [1-4]), which are strong one-shot alternatives with lower computational cost. Including such baselines would provide a fairer evaluation of MIRI’s benefits.
- The method is specifically designed for settings where the missingness is independent of the missing values (MCAR). However, MNAR settings—more common in practice—are not addressed. The authors should explicitly acknowledge early in the paper that MIRI is unlikely to perform well under MNAR conditions, and clarify that its benefits are tied to the MCAR assumption.

### Minor concerns
- Although the Gaussian Mixture with uniform weights can be easily drawn from the context, the equation in L270 is not a valid density without including the weights because it’s unnormalized. I suggest either including the “proportional to” sign or explicitly including the weights.

## References

[1] Nazabal, Alfredo, et al. "Handling incomplete heterogeneous data using vaes." Pattern Recognition 107 (2020): 107501.

[2] Mattei, Pierre-Alexandre, and Jes Frellsen. "MIWAE: Deep generative modelling and imputation of incomplete data sets." International conference on machine learning. PMLR, 2019.

[3] Ma, Chao, et al. "Vaem: a deep generative model for heterogeneous mixed type data." Advances in Neural Information Processing Systems 33 (2020): 11237-11247.

[4] Peis, Ignacio, Chao Ma, and José Miguel Hernández-Lobato. "Missing data imputation and acquisition with deep hierarchical models and hamiltonian monte carlo." Advances in Neural Information Processing Systems 35 (2022): 35839-35851.

---

> ### Author Rebuttal · Authors · 2025-07-30
>
> # Comparison with DiffPuter
> >The proposed method seems closely related to DiffPuter, ... Could the authors elaborate further on the similarities and differences beyond the brief discussion in lines L258–262? A more in-depth comparison could clarify the novelty of MIRI and affect the originality rating.
>
> DiffPuter [1] and MIRI are motivated from different objectives: MIRI minimizes the mutual information between the mask and imputed data while DiffPuter or MICE [2] performs a pseudo likelihood maximization repeatedly where the missing data is replaced with mean-impute (see Section 3.2 in DiffPuter). The ingenuity of DiffPuter is that instead of maximizing the joint likelihood explicitly over both missing and observed variables, it trains a generative model (or implicit likelihood model) on the imputed dataset to produce the mean impute used in the next iteration. This overcomes the difficulties of training and sampling from a high-dimensional likelihood model, which is often intractable and has to be reduced to a dimension-wise regression problem (like MICE).
>
> Both methods train flow-based generative models but use them differently: We use flow model to "match distributions" (as required by Proposition 2) while DiffPuter learns a joint score model to sample conditionally. These two schools of thoughts leads to different choices of flow models.
>
> MIRI chooses rectified flow as it efficiently constructs a flow between two probability distributions. DiffPuter chooses diffusion model as it can use the joint score model to "inpaint" missing variables.
>
> Comparing to DiffPuter, MIRI has two clear advantages:
> - MIRI has a clear information theoretic objective. However, DiffPuter does not maximize a clear utility function (e.g., the marginal likelihood of the observed variables).
> - Inpainting is not a proper conditional sampling procedure which is described in the algorithm in Section 3.2 of [1]. The inpainting procedure undermines the forward-backward process that is the key to the diffusion model. In fact, conditional sampling using diffusion model is non-trivial, and often requires training a dedicated conditional score [3]. However, training a conditional score is difficult in missing data setting as the unobserved variables vary from sample to sample. In our algorithm, the transport between two conditional probabilities using rectified flow is justified through Theorem 1.
>
> [1] Zhang et al., DiffPuter: Empowering Diffusion Models for Missing Data Imputation. ICLR 2025.
>
> [2] van Buuren and Groothuis-Oudshoorn, mice: Multivariate imputation by chained equations in R. J. Stat. Softw., 2011.
>
> [3] Sharrock et al., Sequential Neural Score Estimation: Likelihood-Free Inference with Conditional Score Based Diffusion Models, ICML 2024.
>
> ---
>
> # GAIN Experiment Setup
> >GAIN performs surprisingly poorly in the CIFAR MCAR experiments. Could the authors provide insight into this result? Was the model properly tuned or trained under comparable conditions?
>
> We thank the reviewer for raising this point. For the CIFAR-10 experiments, we adapted the official GAIN implementation [4]. We used a CNN-based generator–discriminator identical to MIRI’s network architecture. To ensure a fair comparison, we applied the same learning rate, batch size, and optimizer settings as used in our method.
> We tested our setting using the MNIST dataset, and our selection of parameters produces convincing handwriting samples which are comparable to Figure 4 in [4].
>
> Our implementation of GAIN can be found at code/competitors/gain_imp.py in the supplementary material.
>
> As we briefly mentioned in Section 2.2, GAN is more unstable than training a normal deep neural network model due to the adversarial training. The parameters like learning rates and number of epochs may have significant impact to the final outcome.
> However, the proposed method requires only optimizes a least squares objective using SGD, thus, is significantly more stable and robust against different choices of tuning parameters (see our ablation study, sensitivity experiments in the **supplementary material**).
>
> [4] Yoon et al., GAIN: Missing Data Imputation using Generative Adversarial Nets. ICML 2018.
>
> ---
>
> # MAR, MNAR
> > However, a more explicit discussion is needed regarding the assumptions behind their method—particularly the reliance on MCAR missingness mechanisms. MIRI is unlikely to perform well in MNAR settings, which are common in real-world data, and this should be clearly stated in the paper. ...
>
> We thank reviewer for raising this excellent point and we will clearly state the limitations of missingness assumption in the revision. Indeed, our method may not work well under MNAR settings as the missing mask may be strongly correlated with the ground truth, making any algorithm that minimizes the mutual information unlikely to work. Interestingly, Section 6 highlights this limitation across a broad range of existing imputation methods. The methods discussed there also implicitly minimize mutual information, making them similarly unsuitable for MNAR scenarios. We will discuss this matter in more detail in the revision.
>
> However, we would also like to highlight that our method not only works with MCAR, but also MAR as a way of enforcing the conditional independence $X_\\mathrm{miss} \\perp M | X_\\mathrm{obs}$.
>
> Please see a proof at the end of the reply.
>
> ---
>
> # VAE Baseline & Additional Results
> >The literature review underrepresents recent progress in deep generative modelling approaches for missing data imputation, particularly VAE-based imputation methods (e.g. [1-4]), which are strong one-shot alternatives with lower computational cost. Including such baselines would provide a fairer evaluation of MIRI’s benefits.
>
> Thank you for raising this point! We have now included MIWAE [5] as an additional baseline.
>
> We compare our method with MIWAE under MCAR, MAR, and MNAR settings. Below table shows the MMD comparisons. We will include this result in the revision.
>
> |Mechanism|$p_{\mathrm{miss}}$|Method|forest|gas|housing|parkinsons|solar|stock|
> |-|-|-|-|-|-|-|-|-|
> |MCAR |0.2|**MIWAE**|0.220 ± 0.042|OOM|0.164 ± 0.010|0.057 ± 0.016|0.157 ± 0.011|0.085 ± 0.007|
> |||**MIRI (Ours\*)**|**0.066 ± 0.014**|**0.045 ± 0.008**|**0.083 ± 0.010**|**0.031 ± 0.007**|**0.076 ± 0.007**|**0.033 ± 0.007**|
> ||0.4|**MIWAE**|0.247 ± 0.018|OOM|0.166 ± 0.016|0.058 ± 0.010|0.165 ± 0.015|0.086 ± 0.014|
> |||**MIRI (Ours\*)**|**0.069 ± 0.010**|**0.044 ± 0.010**|**0.083 ± 0.004**|**0.031 ± 0.007**|**0.080 ± 0.007**|**0.030 ± 0.006**|
> ||0.6|**MIWAE**|0.245 ± 0.018|OOM|0.154 ± 0.014|0.061 ± 0.012|0.159 ± 0.019|0.089 ± 0.011|
> |||**MIRI (Ours\*)**|**0.062 ± 0.012**|**0.043 ± 0.006**|**0.089 ± 0.009**|**0.029 ± 0.009**|**0.077 ± 0.007**|**0.026 ± 0.003**|
> |MAR|0.4†|**MIWAE**|0.313 ± 0.030|0.216 ± 0.006|0.136 ± 0.014|0.091 ± 0.014|0.166 ± 0.048|0.108 ± 0.030|
> |||**MIRI (Ours\*)**|**0.084 ± 0.020**|**0.045 ± 0.008**|**0.121 ± 0.003**|**0.037 ± 0.012**|**0.092 ± 0.021**|**0.037 ± 0.007**|
> |MNAR|0.4|HyperImpute|0.178 ± 0.008|0.049 ± 0.005|**0.087 ± 0.017**|Error|Error|0.069 ± 0.010|
> |||GAIN|0.704 ± 0.005    |0.805 ± 0.002|0.750 ± 0.006|0.197 ± 0.011|0.325 ± 0.004|0.179 ± 0.008|
> |||TabCSDI|0.704 ± 0.006|0.810 ± 0.013|0.739 ± 0.010|0.195 ± 0.013|0.327 ± 0.006|0.190 ± 0.004|
> |||MissDiff|0.085 ± 0.028|0.076 ± 0.015|0.116 ± 0.007|0.049 ± 0.005|0.093 ± 0.014|0.057 ± 0.004|
> |||Sinkhorn|0.259 ± 0.036|0.307 ± 0.019|0.262 ± 0.017|0.089 ± 0.018|0.179 ± 0.027|0.098 ± 0.005|
> |||**MIWAE**|0.220 ± 0.042|OOM|0.186 ± 0.012|0.071 ± 0.017|0.146 ± 0.027|0.087 ± 0.002|
> |||TDM|0.284 ± 0.056|0.176 ± 0.006|0.209 ± 0.007|0.098 ± 0.020|0.171 ± 0.042|0.111 ± 0.001|
> |||**MIRI (Ours\*)**|**0.066 ± 0.014**|**0.045 ± 0.008**|0.097 ± 0.002|**0.043 ± 0.007**|**0.093 ± 0.003**|**0.052 ± 0.002**|
>
> †MAR: 50% of observed variables are used to generate mask.
>
> We also provide additional results on MAR and MNAR data. Please see our response to the Reviewer k1XP.
>
> [5] Mattei and Frellsen, MIWAE: Deep Generative Modelling and Imputation of Incomplete Data Sets. ICML 2019.
>
> ---
>
> # Proof for MAR Setting
>
> Define the notation:  $X_\mathrm{obs} = X_M$ and $X_\mathrm{miss} = X_{1-M}$, and consider the goal of minimizing the **conditional mutual information**:
>
> $$\mathrm{I}[X_\mathrm{miss}(g); M | X_\mathrm{obs}] = E_{(x,m)\sim (X,M)}\log \frac{p_g(x_{1-m}, m |x_m )}{p(x_{1-m}| x_m)p(m|x_m)}$$
>
> Following a very similar proof in proposition 1, we have the iterative algorithms to reduce the mutual information. At iteration $t$:
>
> - $$
>   g^* = \arg\min_{g} E_{(x,m) \sim (X,M)}\left[\log \frac{p(x_{1-m}, m \mid x_m )}{q(x_{1-m} \mid x_m) q(m \mid x_m)}\right],
>   $$
>   where $p$ is the density function of $P_{X^{(t)}, M}$ and $q$ is the density function of $P_{X^{(t-1)}}$, the probability of $X$ in the previous iteration.
>
> - Impute using $g^*$ and repeat.
>
> We now show that  $p(x_{1-m} \mid m, x_m) = q(x_{1-m} \mid x_m)$  is a sufficient condition for $g$ to be optimal in the first step.
>
> Analogous to Proposition 2, we can rewrite the objective function:
>
> \\[
> \begin{aligned}
> E_{(x,m) \sim (X,M)}\left[\log \frac{p(x_{1-m}, m \mid x_m )}{q(x_{1-m} \mid x_m) p(m \mid x_m)}\right] =&E_{(x,m) \sim (X,M)}\left[\log \frac{p(x_{1-m} \mid m, x_m )}{q(x_{1-m} \mid x_m)}\right] \\\\
> =&E_{(m, x_m) \sim (M, X_M)} E_{x_{1-m} \sim X_{1-M} \mid M, X_M} \left[\log \frac{p(x_{1-m} \mid m, x_m )}{q(x_{1-m} \mid x_m)}\right]
> \end{aligned}
> \\]
>
> Note that $p(m \mid x_m)$ does not change across iterations, so we assume $p(m \mid x_m) = q(m \mid x_m)$.
> The inner expectation is a KL divergence between $p_g(x_{1-m} \mid m, x_m)$ and $q(x_{1-m} \mid x_m)$,  which is minimized (to zero) when the two distributions are equal:
>
> \\[
> p(x_{1-m} \mid m, x_m) = q(x_{1-m} \mid x_m)
> \\]
>
> This is exactly the sufficient condition stated in Proposition 2.

---

> > ### Comment · Reviewer_Dwu5 · 2025-08-01
> > **Rebuttal acknowledgement**
> >
> > Thank you for the detailed and thoughtful rebuttal. I appreciate the effort in clarifying the technical aspects, particularly the challenges of conditional sampling in diffusion models and the theoretical justification of your approach. I also value the additional results comparing against MIWAE, which strengthen the empirical evaluation.
> >
> > Your responses have addressed most of my concerns, and I am now more confident in the paper’s contributions. That said, I still believe the manuscript would benefit from a clearer motivation and improved presentation in the Introduction section.

---

> > > ### Author Response · Authors · 2025-08-01
> > > **Thanks for your response!**
> > >
> > > We also thank reviewer for their insightful review. In particular, the comment on the applicability of MIRI on MAR/MNAR data has inspired us to further explore and clarify the method’s theoretical assumptions and motivation.
> > >
> > > Our method is mainly motivated by two factors:
> > >
> > > 1. providing a unified imputation framework for MCAR/MAR data from information theoretic perspective,
> > > 2. leveraging flow-based models to provide high-quality image imputation.
> > >
> > > We will clarify these motivations and improve the presentation of the introduction in the revised paper.
> > >
> > > If the reviewer has any additional question/concern during the discussion period, we will be happy to address them.

---

> > > > ### Comment · Reviewer_Dwu5 · 2025-08-07
> > > > **Final comment**
> > > >
> > > > Dear authors,
> > > >
> > > > Thank you for your responsiveness during the rebuttal. While I cannot fully assess the updated Introduction, I trust that, with the valuable feedback provided by the reviewers, both the clarity and motivation of the paper will improve in the camera-ready version. All my other concerns have been addressed. I will increase my score and recommend acceptance.

---

> > > > > ### Author Response · Authors · 2025-08-08
> > > > > **Thank you!**
> > > > >
> > > > > We thank the reviewer for encouraging and constructive feedback! We will definitely clarify our motivation and incorporate our rebuttal to all reviewers into the revision of our paper!

---

### Official Review · Reviewer_iwML · 2025-07-02

**Clarity:** 2
**Significance:** 2
**Originality:** 2
**Rating:** 4
**Confidence:** 3

**Summary:**

This paper introduces a new iterative approach for missing data imputation that progressively reduces the statistical dependence between imputed values and the observed mask. At each iteration, a rectified flow model is trained to transform the current imputation distribution toward a reference distribution where imputed values are statistically independent of the mask.

**Questions:**

1. How many iterations are needed for the proposed method to converge?

2. What are the model configuration and training details of the rectified flow model?

3. What will be the performance for other missing patterns like Missing at random(MAR) and Missing Not At Random (MNAR)?

**Ethical Concerns:**

["NO or VERY MINOR ethics concerns only"]

**Final Justification:**

The authors have addressed the concerns regarding the motivation for using rectified flow and the long running time. One remaining issue is the moderate performance on entry-wise metrics such as RMSE. Although the authors claim that the method is primarily designed for distributional metrics. This method can be adapted to entry-wise metrics through repeated sampling and expectation estimation. However, the paper lacks experiments to discuss this matter. Therefore, I am increasing my score to borderline accept.

**Limitations:**

yes

**Quality:**

2

**Strengths And Weaknesses:**

**Strengths**

1. The core idea of iteratively reducing dependence between the imputation and the mask with a generative model is novel.

2. The paper is in general well-written and structured.

3. The method is theoretically grounded.

4. The method shows better performance on distribution level metric MMD compared to previous SOTA methods like Hyperimpute.

**Weakness**

1. High computational cost:
The proposed method requires training a new rectified flow model and performing sampling at each iteration, which can be computationally expensive. However, the paper does not report imputation time or compare runtime performance with other baselines. A systematic evaluation of efficiency would be helpful for understanding the method’s practicality.

2. Motivation for Choosing Rectified Flow Not Justified:
While rectified flow is used as the generative backbone at each step, the paper does not clearly explain why it is preferable to alternatives like diffusion models. Nor is there any ablation to validate this choice empirically. A deeper justification—either theoretical or experimental—would strengthen the case for this design decision.

3. Underperformance on Entry-Wise Metrics:
For standard entry-wise imputation metrics such as RMSE and MAE, the evaluation is only conducted on synthetic data. On this task, the proposed method performs worse than existing baselines like HyperImpute, raising concerns about its overall performance.

4. Incomplete Baseline Comparisons:
The experimental section lacks several related baselines, e.g. MOT [1], TDM [2], which are both based on distribution matching.

[1]  Muzellec, Boris, et al. "Missing data imputation using optimal transport." International Conference on Machine Learning. PMLR, 2020.

[2]  Zhao, He, et al. "Transformed distribution matching for missing value imputation." International Conference on Machine Learning. PMLR, 2023.

---

> ### Author Rebuttal · Authors · 2025-07-30
>
> # Computational Cost
> > ... the paper does not report imputation time or compare runtime performance with other baselines. A systematic evaluation of efficiency would be helpful for understanding the method’s practicality.
>
> We appreciate that computational cost is crucial for high-dimensional data imputation. However, we would like to highlight that it is decisively faster than dimension-wise method in high dimensional settings. Hyperimpute or MICE does not scale well in high-dimensional settings, due to their round-robin algorithm. Moreover, the “sequential” nature of these algorithms, making parallelization difficult.
>
> In comparison, the proposed method is "vector in, vector out", which can fully utilize a variety of hardware which are specialized in dealing with high-dimensional structured data (e.g., images).
>
> On CIFAR dataset, Hyperimpute failed to finish imputation in a preset time-limit (24 hours), so is not included in the final result, while MIRI finishes in around 3.5 hours on the same environment (Pytorch with CUDA).
>
> We would also like to highlight that, DiffPuter [3], a state-of-the-art imputation algorithm also adopts a similar algorithm (training and sampling from a flow-based model repeated), which will have a very similar computational cost as our method.
>
> Below, we compare the average computation time (hours) of 1000 samples with various dimensions. We include methods based on diffusion (DiffPuter, TabCSDI), round-robin (HyperImpute), distribution matching (MOT [1], TDM [2]) and VAE (MIWAE [4]). It can be seen that MIRI is on par with all other diffusion-based methods, if not faster. It is way faster than HyperImpute in high-dimensional space.
>
> | **Methods**| **50d**| **200d**| **500d**| **1000d**| **2000d**| **5000d**|
> |-|-|-|-|-|-|-|
> | HyperImpute| 0.08±0.03| 0.23±0.02|1.18±0.51|3.84±0.99| 11.65±3.34| 57.54±0.00|
> | TabCSDI| 0.18±0.00| 0.20±0.01| 0.26±0.00|0.36±0.00| 0.53±0.01| 1.10±0.01|
> | DiffPuter| 2.30±0.01| 3.22±0.18| 5.06±0.01|7.70±0.05| 12.40±0.21| 29.10±0.00|
> | MIWAE| 0.11±0.00| 0.30±0.00|0.90±0.03|2.02±0.01| OOM| OOM|
> | MOT | 0.31±0.00| 0.33±0.00|0.36±0.00|0.38±0.00| 0.49±0.02| 0.56±0.00|
> | TDM | 0.32±0.00| 0.55±0.00|1.30±0.02|3.30±0.04| 13.90±0.83| OOM|
> | **MIRI (Ours\*)** | 0.17±0.00|0.21±0.00|0.28±0.02| 0.37±0.02| 0.59±0.02| 1.30±0.01|
>
> [1] Muzellec et al., Missing data imputation using optimal transport. ICML 2020.
>
> [2] Zhao et al.,transformed distribution matching for missing value imputation. ICML 2023.
>
> [3] Zhang et al., DiffPuter: Empowering Diffusion Models for Missing Data Imputation. ICLR 2025.
>
> [4] Mattei and Frellsen, MIWAE: Deep Generative Modelling and Imputation of Incomplete Data Sets. ICML 2019.
>
> ---
>
> # Justification of Rectified Flow
> > the paper does not clearly explain why it is preferable to alternatives like diffusion models ... A deeper justification—either theoretical or experimental—would strengthen the case for this design decision.
>
> Thanks for raising this point. We would like to clarify that diffusion model **cannot** be used, as the initial distribution of the backward process must be a Gaussian distribution. However, Proposition 2 requires us to transport from a specific conditional distribution $p$ to $q$, (eq. 5), where $p$ is almost always not a Gaussian distribution. Rectified flow is an example of the Flow Matching (Lipman et al., 2023), which are designed to transport between two distributions. We choose rectified flow for its simplicity both in theory and practice.
>
> In short, diffusion model is designed to "sample", while rectified flow is designed to "transport".
>
> More specifically, we also utilized a mathematical property of Rectified Flow, called Marginal Preserving Property (Theorem 3.3 in Liu et al., 2023) to prove Theorem 1 (Lemma 1) where we have shown that sampling using the vector field in eq. 6 can indeed transport samples from $p$ of $q$. We will clarify this point in the revision and discuss the rationale of our choice.
>
> Lipman et al., Flow Matching for Generative Modeling, ICLR, 2023
>
> ---
>
> # Underperformance on Entry-Wise Metrics
> > For standard entry-wise imputation metrics such as RMSE and MAE, the evaluation is only conducted on synthetic data. On this task, the proposed method performs worse than existing baselines like HyperImpute, raising concerns about its overall performance.
>
> Thanks for pointing it out! There are two types of imputation: Mean imputation (HyperImpute, DiffPuter) and distributional imputation (MIRI, GAIN).
>
> The former imputes data such that  $X_\mathrm{imp} = E[X_{\mathrm{miss}} \mid X_{\mathrm{obs}}]$,  while the latter imputes data such that  $X_\mathrm{imp} \sim P(X_{\mathrm{miss}} \mid X_\mathrm{obs})$.  Distributional imputation is harder than mean imputation, as we can draw multiple imputations from  $P(X_{\mathrm{miss}} \mid X_\mathrm{obs})$  and compute  $E[X_{\mathrm{miss}} \mid X_{\mathrm{obs}}]$,  but not vice versa. Our method is inspired by GAIN, and therefore also focuses on **distributional imputation**.
>
> The key in distributional imputation is to test if the impute data looks like it is drawn from the probability distribution that generated the ground truth. For this reason, we test the statistical discrepancy between imputed data and the ground truth using a divergence measure such as MMD.
>
> Moreover, we can see that Mean impute is optimized for the means squared error metric as $E[X_{\mathrm{miss}}|X_{\mathrm{obs}}] = \arg\min_V E[\|X_{\mathrm{miss}} - V\|^2|X_{\mathrm{obs}}]$, so it is not surprising that methods like Hyperimpute achieves a slightly better MSE.
>
> We also want to highlight that, although our method is not designed to minimize MSE, it also performs well on entry-wise metrics based on MSE such as PSNR in the CIFAR experiment (Table 1):
> \begin{equation}
> \text{PSNR} = 10 \cdot \log_{10} \left( \frac{L^2}{\text{MSE}} \right),
> \end{equation}
> This indicates though MIRI can still achieve superior entry-wise performance on **high dimensional image dataset**, especially when the missing rate is high. The PSNR metric is widely used in inpainting (which is similar to our setting), like in Elharrouss et al. 2020.
>
> We will add this explanation to the revision.
>
> We also tested MIRI under multiple imputations in synthetic data used in Section 6.1 ($p_{\mathrm{miss}}=0.6$), which can effectively reduce RMSE:
>
> |#Mult. Imp.|MCAR|MAR|MNAR |
> |---|---|---|---|
> |1|1.816|1.047|2.167|
> |5|1.562|0.898|1.700|
> |**10**|**1.513**|**0.866**|**1.666**|
>
> Elharrouss et al., Image Inpainting: A Review. Neural Process. Lett., 2020.
>
> ---
>
> # MAR, MNAR
> > What will be the performance for other missing patterns like Missing at random (MAR) and Missing Not At Random (MNAR)?
>
> We have done theoretical analysis and empirical evaluation on MAR data.
>
> Theocratically, our method is also justified for the MAR setting. Our method is motivated by the key assumption of MCAR, that true data is independent of the mask.
>
> In the MAR setting, the independent assumption becomes $X_{\mathrm{miss}} \perp M | X_{\mathrm{obs}}$. Following the same rationale, we can minimize the **conditional mutual information** $\mathrm{I}[X_\mathrm{miss}(g); M | X_\mathrm{obs}]$ given the observed variables.
>
> We prove that the condition in Proposition 2 is also the sufficient condition of minimizing the conditional mutual information. In other words, running the unmodified MIRI on MAR data will reduce the conditional mutual information $\mathrm{I}$ iteratively.
>
> We provide a proof at the end.
>
> In addition to these new theoretical results, we also provide empirical proofs that our method works on MAR data. Please see our response to the Reviewer k1XP.
>
> We will add new empirical results and theoretical results to the revised paper.
>
> ---
>
> # Experiment Settings, Missing Baseline
>
> We perform 10 iterations of the MIRI algorithm, which we observe is long enough for the mutual information to converge. We will clarify this in the revision.
>
> Please see the Appendix E and supplementary materials (`hyperparameters.md`) for model configuration and training details.
>
> We also included two methods MOT and TDM as baselines in MAR and MNAR experiments, please see our response to reviewer k1XP.
>
> ---
>
> # Proof for MAR Setting
>
> Define the notation:  $X_\mathrm{obs} = X_M$ and $X_\mathrm{miss} = X_{1-M}$, and consider the goal of minimizing the **conditional mutual information**:
>
> $$\mathrm{I}[X_\mathrm{miss}(g); M | X_\mathrm{obs}] = E_{(x,m)\sim (X,M)}\log \frac{p_g(x_{1-m}, m |x_m )}{p(x_{1-m}| x_m)p(m|x_m)}$$
>
> Following the a very similar proof in proposition 1, we have the iterative algorithms to reduce the mutual information. At iteration $t$:
>
> - $$
>   g^* = \arg\min_{g} E_{(x,m) \sim (X,M)}\left[\log \frac{p(x_{1-m}, m \mid x_m )}{q(x_{1-m} \mid x_m) q(m \mid x_m)}\right],
>   $$
>   where $p$ is the density function of $P_{X^{(t)}, M}$ and $q$ is the density function of $P_{X^{(t-1)}}$, the probability of $X$ in the previous iteration.
>
> - Impute using $g^*$ and repeat.
>
> We now show that  $p(x_{1-m} \mid m, x_m) = q(x_{1-m} \mid x_m)$  is a sufficient condition for $g$ to be optimal in the first step.
>
> Analogous to Proposition 2, we can rewrite the objective function:
>
> \\[
> \begin{aligned}
> E_{(x,m) \sim (X,M)}\left[\log \frac{p(x_{1-m}, m \mid x_m )}{q(x_{1-m} \mid x_m) p(m \mid x_m)}\right] =&E_{(x,m) \sim (X,M)}\left[\log \frac{p(x_{1-m} \mid m, x_m )}{q(x_{1-m} \mid x_m)}\right] \\\\
> =&E_{(m, x_m) \sim (M, X_M)} E_{x_{1-m} \sim X_{1-M} \mid M, X_M} \left[\log \frac{p(x_{1-m} \mid m, x_m )}{q(x_{1-m} \mid x_m)}\right]
> \end{aligned}
> \\]
>
> Note that $p(m \mid x_m)$ does not change across iterations, so we assume $p(m \mid x_m) = q(m \mid x_m)$.
> The inner expectation is a KL divergence between $p_g(x_{1-m} \mid m, x_m)$ and $q(x_{1-m} \mid x_m)$,  which is minimized (to zero) when the two distributions are equal:
>
> \\[
> p(x_{1-m} \mid m, x_m) = q(x_{1-m} \mid x_m)
> \\]
>
> This is exactly the sufficient condition stated in Proposition 2.

---

> > ### Comment · Reviewer_iwML · 2025-08-01
> >
> > I thank the authors for their detailed clarification, which has resolved my primary concerns. The new experiments convincingly demonstrate that the proposed method outperforms distribution-matching approaches like MOT and TDM and trains significantly faster than previous iterative methods.
> >
> > I have one remaining question: the proposed method is reported to be significantly faster than DiffPuter. Could the authors elaborate on the key factors that contribute to this speed advantage?

---

> > > ### Author Response · Authors · 2025-08-02
> > > **Thanks for your Comments!**
> > >
> > > Thanks for your insightful comments!
> > >
> > > The speed difference between **MIRI** and **DiffPuter** is due to the fact that DiffPuter must run the diffusion process multiple times. To approximate the expectation in the E-step*, we must draw multiple samples and average them, which means solving the sampling SDE N times. In the official implementation of DiffPuter, N is set to 20 by default.
> > >
> > > By contrast, MIRI solves a single ODE per iteration to transport samples from $p$ to $q$ (Algorithm 1, line 10), making it significantly faster.
> > >
> > > This also clarifies why, when dimension = 1000, 2000, 5000, the computation time of DiffPuter is roughly 20 times that of MIRI.
> > >
> > > ---
> > >
> > > *Please see "Expectation Step", Section 3.2 in Zhang et al., 2025.
> > >
> > > Zhang et al., Empowering Diffusion Models for Missing Data Imputation, ICLR 2025.

---

> > > > ### Comment · Reviewer_iwML · 2025-08-04
> > > >
> > > > That makes sense, thanks for the clarification!

---

> > > > > ### Author Response · Authors · 2025-08-04
> > > > > **Thank you!**
> > > > >
> > > > > We thank the reviewer for insightful comments and time and effort dedicated on reviewing our paper! If any further question arises, please do not hesitate to comment and we are happy to respond during the discussion period.

---

### Note · Authors · 2025-08-12

We thank all reviewers for the thoughtful questions and comments that have made our work stronger. Below we highlight our key response to reviewers’ comments regarding the motivation and the applicability of the proposed method:

Our approach is motivated by **the independence assumption in MCAR and MAR settings**. In the revision, we will **make this assumption explicit**, provide a formal justification for the MAR case (proof included in the rebuttal) and add new empirical evaluations on MAR and MNAR data. For the MNAR case, we will also discuss a potential extension of MIRI via a latent-variable formulation (see our response to Reviewer yRf4).

To address comments on time-series data imputation, we will incorporate results from a self-supervised variant of MIRI, which demonstrates strong RMSE performance.

We will also integrate baseline methods suggested by reviewers (e.g., MOT, TDM and MIWAE), provide detailed computational cost comparison across all methods and clarify the performance metrics (RMSE vs. MMD) in the revision.

Our paper introduces a novel, unifying framework of missing data imputation, by minimizing an information theoretic objective (mutual information). Leveraging state-of-the-art flow-based generative models, we achieve superior imputation performance on high-dimensional image data. Our framework also suggests a general, flow-based approach to minimize mutual information that may have impact beyond missing data imputation.

We thank reviewers for encouraging comments which have definitely strengthened our paper, and will revise our paper to address your comments. Thank you once again for your time and consideration.

---

### Decision · Program_Chairs · 2025-09-17

**Decision:**

Accept (poster)

**Comment:**

The authors introduced novel method for data imputation by minimizing the mutual information between the imputed data and the missing mask. This proposed technique has solid theoretical foundation. Furthermore, the empirical evidence was substantially enhanced throughout the rebuttal period. One limitation is that the proposed approach lacks a theoretical guarantee in MNAR scenarios; however, the authors still present empirical evidence for these settings. I highly recommend that the authors incorporate all the discussions during the rebuttal into the final camera-ready version of their manuscript.